# Spontaneous hydrolysis and spurious metabolic properties of α-ketoglutarate esters

Seth J. Parker [1,2,6✉], Joel Encarnación-Rosado[1,2], Kate E. R. Hollinshead[1,2,7], David M. Hollinshead[3,7], Leonard J. Ash[4], Juan A. K. Rossi[2], Elaine Y. Lin[1,2], Albert S. W. Sohn[1,2], Mark R. Philips [2], Drew R. Jones[4,5] & Alec C. Kimmelman[1,2✉]

α-ketoglutarate (KG), also referred to as 2-oxoglutarate, is a key intermediate of cellular metabolism with pleiotropic functions. Cell-permeable esterified analogs are widely used to study how KG fuels bioenergetic and amino acid metabolism and DNA, RNA, and protein hydroxylation reactions, as cellular membranes are thought to be impermeable to KG. Here we show that esterified KG analogs rapidly hydrolyze in aqueous media, yielding KG that, in contrast to prevailing assumptions, imports into many cell lines. Esterified KG analogs exhibit spurious KG-independent effects on cellular metabolism, including extracellular acidification, arising from rapid hydrolysis and de-protonation of α-ketoesters, and significant analog-specific inhibitory effects on glycolysis or mitochondrial respiration. We observe that imported KG decarboxylates to succinate in the cytosol and contributes minimally to mitochondrial metabolism in many cell lines cultured in normal conditions. These findings demonstrate that nuclear and cytosolic KG-dependent reactions may derive KG from functionally distinct subcellular pools and sources.

[1] Department of Radiation Oncology, New York University School of Medicine, New York, NY, USA. [2] Perlmutter Cancer Center, New York University School of Medicine, New York, NY, USA. [3] Elixir Software Ltd., Macclesfield, Cheshire, UK. [4] Division of Advanced Research Technologies, New York University School of Medicine, New York, NY, USA. [5] Department of Biochemistry and Molecular Pharmacology, New York University School of Medicine, New York, NY, USA. [6] Present address: Department of Biochemistry & Molecular Biology, University of British Columbia, Vancouver, BC, Canada. [7] These authors contributed equally: Kate E. R. Hollinshead, David M. Hollinshead. ✉email: seth.parker@bcchr.ca; alec.kimmelman@nyulangone.org

L ipid bilayers consist of a hydrocarbon core that limits passive diffusion of polar, charged, and/or hydrophilic solutes into cells; including organic acids, sugars, and ions[1]. The low permeability of glucose, urea, monovalent ions, and other polar metabolites, ranging from $10^{-7}$ to $10^{-13}$ cm/s, necessitates the engagement of complex transport systems to facilitate the diffusion of many metabolites across cellular membranes to support metabolic activity and homeostasis[2–5]. Optimizing membrane partitioning of drug-like molecules is a major challenge for therapeutic development to obtain desirable pharmacokinetic properties[6]. Esterification is one such approach that can improve drug permeability by increasing lipophilic properties. Esterified analogs are designed to act as prodrugs that hydrolyze within the cell by non-specific esterases, such as acetyl- and carboxyl-esterases, releasing the active molecule[7,8]. Esterification is also applied broadly outside of the pharmaceutical space as a technique to improve the membrane permeability of hydrophilic metabolites and polar molecules. For example, fluorescent dyes (e.g., sulfonated cyanine Cy3 and Cy5, fluorescein) are poorly permeable and require either microinjection or esterification (e.g., fluorescein diacetate) to improve delivery within live cells[9]. From a metabolism perspective, esterification is used to bypass transport-mediated import or deliver metabolites thought to be impermeable to cells. For example, α-ketoglutarate (KG) is reportedly impermeable to cell membranes and esterified analogs, such as dimethyl-α-ketoglutarate (DMKG) and octyl-α-ketoglutarate, are used to supplement cells with KG to study how it fuels central carbon metabolism and epigenetics and protein hydroxylation reactions[10–31].

Here, we show that esterified KG analogs rapidly hydrolyze to α-ketoglutaric acid in aqueous conditions independently of cellular esterases. Given its low pKa, α-ketoglutaric acid rapidly released from neutral esterified analogs contributes to the acidification of extracellular media and significant metabolic dysfunction, including reduced glycolytic, respiratory, and/or proliferative potential depending on the KG analog. Hydrolysis was also observed for other metabolite esters; including alanine and fumarate; with varying rates. DMKG followed a distinct hydrolysis mechanism, whereby rapid hydrolysis of the proximal α-ketoester occurred much more rapidly than the distal ester and produced a stable mono-methylated KG that persisted for days and was taken up by cells. More hydrophobic analogs (e.g., tert-butyl alanine) exhibited significant hydrolytic resistance over several days compared to methyl- and ethyl- analogs, suggesting that cell-permeable properties of esterified analogs may be prolonged with large aliphatic modifications. Notably, KG in its conjugate base form, derived from the disodium salt, was imported by many cell lines and did not illicit similar metabolic effects as esterified KG. Imported KG was used predominantly for cytosolic/nuclear dioxygenase activity and minimally contributed to mitochondrial TCA cycle metabolism, suggesting that mitochondrial KG is derived from a functionally distinct source. However, DMKG was able to significantly contribute carbon to mitochondrial TCA cycle metabolism, suggesting that di- or mono-methylated KG retains plasma and mitochondrial membrane permeable properties. Our results highlight significant off-target effects associated with esterified α-ketone metabolites, such as pyruvate and KG, and suggests that caution be used when interpreting metabolic phenotypes associated with these analogs.

## Results

### KG esters hydrolyze spontaneously

The most commonly used cell-permeable analogs of KG include mono- and di- esters such as dimethyl-α-ketoglutarate (DMKG) and octyl-α-ketoglutarate (1-octyl-KG), although other analogs such as (trifluoromethyl)

benzyl α-ketoglutarate (TFMB-KG), benzyl α-ketoglutarate, and trifluorobenzyl α-ketoglutarate ethyl ester (ETaKG) have been used[10–31]. Once across cellular membranes, ester hydrolysis releases α-ketoglutaric acid and an alcohol (e.g., methanol, octanol), the latter of which is removed from cells by passive diffusion and evaporation or metabolism (Fig. 1a)[32]. Previous studies have identified that other esterified metabolite analogs, including dimethyl-itaconate, 4-octyl-itaconate, and dimethyl-oxalylglycine (DMOG), can hydrolyze spontaneously in aqueous conditions and are taken up as either their metabolite and/or mono-ester forms (e.g., methyl-oxalylglycine)[33,34]. To understand if KG esters are also susceptible to spontaneous hydrolysis, we conducted a cell-free hydrolysis assay in DMEM at 37 °C and quenched at specified time points in anhydrous methanol containing an isotopically labeled $^2H_6$-labeled KG standard for quantitation by mass spectrometry. After 72 h, the majority (>50%) of DMKG or 1-octyl-KG were detected as KG suggesting that double or single hydrolysis, respectively, can occur spontaneously and independently of cellular esterases (Fig. 1a, b). Hydrolysis kinetics were distinct between DMKG and 1-octyl-KG. KG derived from complete mono-hydrolysis of 1-octyl-KG was observed after 24 h, whereas a gradual increase in KG release from double hydrolysis of DMKG was observed over the 72-h time course (Fig. 1c). Hydrolysis of 1-octyl-KG occurred as early as 5 m in follow-up short-term assays (Supplementary Fig. 1a). While it has been suggested that KG is impermeable to cellular membranes, the majority of studies exclude KG as a control and rely exclusively on esterified analogs[10,13–19,21–27,29,31]. We hypothesized that the spontaneous hydrolysis of KG esters may affect cell-permeable properties assumed for these more hydrophobic analogs, and KG itself may readily import into cells. To explore this, we cultured a panel of pancreatic cancer cell lines with either methyl acetate (vehicle), KG, or DMKG for 24 h. In all cell lines tested, we observed a significant accumulation of KG when cells were supplemented with either KG or DMKG, suggesting that KG can diffuse into cells (Fig. 1d). KG accumulated to similar extents in KG- or DMKG- treated conditions with considerable heterogeneity across cell lines, suggesting that specific transporter(s) may be required to facilitate the import of KG (Fig. 1d, e). To understand if extracellular KG could assimilate into intracellular metabolite pools, we conducted a reverse $^{13}C$-labeling experiment whereby intracellular KG, synthesized predominantly from glutamine and subsequent deamination of glutamate, was labeled using $^{13}C_5$-glutamine prior to the addition of unlabeled ($^{12}C$) KG or DMKG (Fig. 1e). As expected, glutamine represented the major carbon source for synthesized KG contributing ~75–92% of carbon across cell lines as quantified by the mole percent enrichment (MPE) (Fig. 1f, g). After 24 h, we observed a significant dilution of labeled KG species (M1-M5) by unlabeled (M0) in 8988S, AsPC1, and 8988T cell lines arising from import and assimilation into intracellular KG pools (Fig. 1f; Supplementary Fig. 1b). Extracellular KG or DMKG comprised ~40–80% of the intracellular pool, depending on the cell line, suggesting that uptake can increase cellular levels (Fig. 1d) and provide a major source of the intracellular KG (Fig. 1g).

### Hydrolysis is common to other metabolite esters

Esterification provides more than improved drug membrane penetrance and can be used to select for desired therapeutic activity caused by analogs themselves. For example, dimethyl-fumarate (DMF); combined with the $Mg^{2+}$, $Ca^{2+}$, and $Zn^{2+}$ salts of monoethyl-fumarate (MEF); is currently approved in Germany as an oral treatment for psoriasis marketed under the tradename Fumaderm[35]. In the United States, the FDA has approved DMF (Tecfidera), as well as mono-methyl fumarate (MMF; Bafiertam)

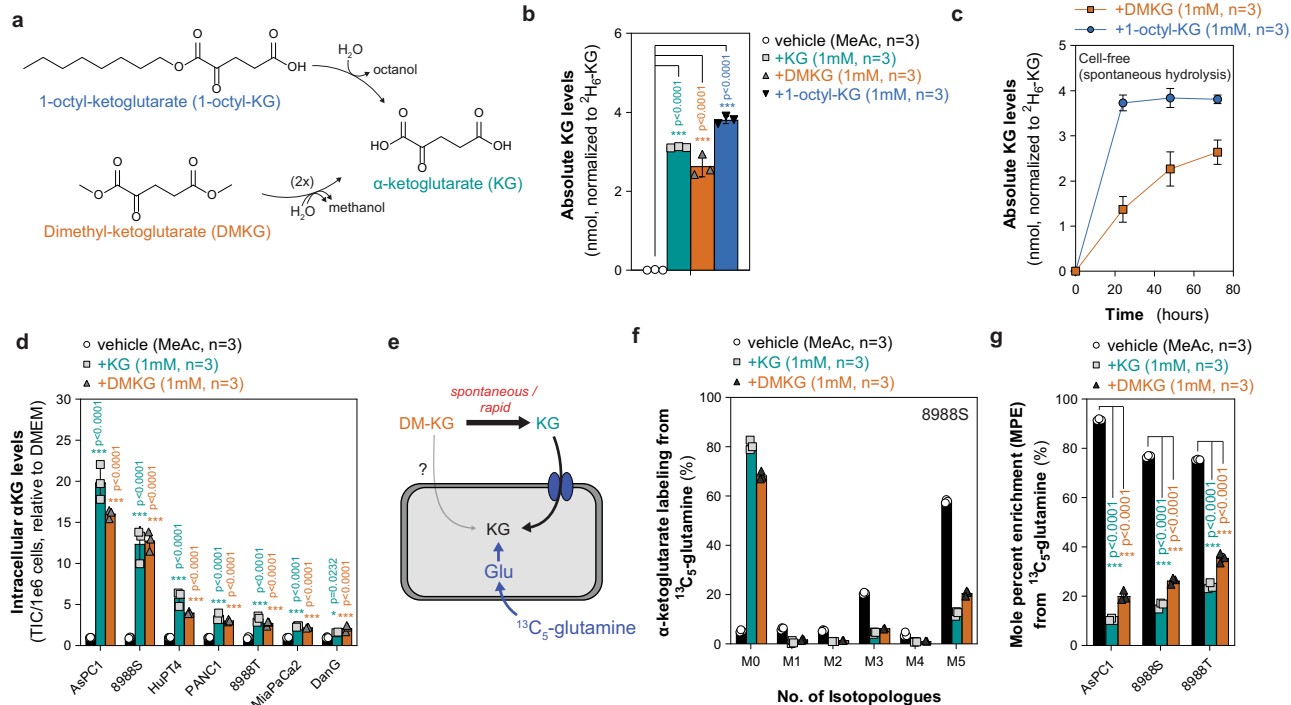

**Fig. 1 Spontaneous hydrolysis of α-ketoglutarate esters and equivalent uptake of non-esterified α-ketoglutarate. a** Schematic depicting hydrolysis of dimethyl-α-ketoglutarate (DMKG) and 1-octyl-α-ketoglutarate (1-octyl-KG) to α-ketoglutarate (KG). **b** Absolute KG levels quantified after cell-free hydrolysis of 1 mM of DMKG or 1-octyl-KG after 72 h compared to vehicle (methyl acetate; MeAc) and 1 mM KG. Data are mean ± s.d, $n = 3$ biologically independent experiments; significance determined by one-way ANOVA using Dunnett's multiple comparisons test, ***$p < 0.0001$. **c** Time course of absolute KG levels quantified after cell-free hydrolysis of DMKG and 1-octyl-KG over 72 h. Data are mean ± s.d., $n = 3$ biologically independent experiments. **d** Intracellular KG levels in AsPC1, 8988S, HuPT4, PANC1, 8988T, MiaPaCa2, and DanG cells treated with methyl acetate or 1 mM of KG or DMKG for 24 h. Data are plotted as relative to vehicle-treated condition; mean ± s.d., $n = 3$ biologically independent samples; significance determined by two-way ANOVA using Dunnett's multiple comparisons test, *$p < 0.05$, ***$p < 0.0001$. **e** Schematic depicting stable-isotope dilution experiment using $^{13}C_5$-glutamine to label intracellular KG and quantify KG or DMKG contribution to the intracellular pool of KG. **f** KG mass isotopologue distribution of 8988S cells labeled with $^{13}C_5$-glutamine and treated with either methyl acetate or 1 mM of KG or DMKG for 24 h. Data are mean ± s.d., $n = 3$ biologically independent samples. **g** Mole percent enrichment (MPE) of intracellular KG labeling of AsPC1, 8988S, and 8988T cells cultured with $^{13}C_5$-labeled glutamine and treated with methyl acetate or 1 mM of unlabeled ($^{12}C$) KG or DMKG for 24 h. Data are mean ± s.d., $n = 3$ biologically independent samples; significance determined by two-way ANOVA using Dunnett's multiple comparisons test, ***$p < 0.0001$.

and diroximel fumarate (Vumerity), as a treatment for relapsing-remitting multiple sclerosis (MS). It has been proposed that DMF acts as a pro-drug and that MMF, but not fumarate, provides the majority of immuno-modulatory activity for both psoriasis and relapsed MS, although this is debated[36]. In pharmacokinetic studies, DMF rapidly hydrolyzes to MMF in circulation, and subsequent hydrolysis of MMF to fumarate followed by its metabolism has been proposed[37–39]. To explore the hydrolysis kinetics of fumarate esters, we conducted cell-free hydrolysis experiments using DMF as well as diethyl-fumarate (DEF). In agreement with previous studies, both DMF and DEF underwent spontaneous double hydrolysis to fumarate in aqueous conditions over 72 h (Fig. 2a). Although rapid hydrolysis of di- to mono-esterified fumarate is expected, we failed to identify a peak for MMF or MEF using our GC-MS analysis; however, gas chromatography may not be suitable to detect MMF and MEF as HPLC was used to quantify levels in pharmacokinetic studies[38,39]. We hypothesized that DMKG may also exhibit a similar hydrolysis mechanism as DMF. Indeed, we observed a significant unidentified peak in our GC-MS analysis of KG ester hydrolysis at 24 h that was unique to DMKG samples (Supplementary Fig. 2a). From the mass spectrum, we identified a putative methoxime-tertbutyldimethylsilyl derivative of mono-methyl α-ketoglutarate (MKG) with a main fragment ion of 246 $m/z$ corresponding to [M-57] (-$C_4H_9$, tert-butyl) (Supplementary

Fig. 2b). Integration of this peak revealed that DMKG rapidly hydrolyzes to the MKG within 24 h, and as early as 5 m, followed by relatively slow hydrolysis to KG, agreeing with our previous measurements of KG release from DMKG (Fig. 2b; Supplementary Fig. 2c). Surprisingly, hydrolysis of DMKG to MKG and subsequent hydrolysis of MKG to KG followed distinct kinetics, suggesting that MKG may be transiently abundant in aqueous conditions similar to MMF (Fig. 2b). To confirm the presence of mono-esterified KG and quantify regioselective hydrolysis of the 1-carbon or 5-carbon ester groups, we conducted a high-resolution liquid chromatography-mass spectrometry (LC-MS) analysis of 8988T cells treated with methyl acetate or 1 mM of KG or DMKG for 3 h. We identified two features in negative mode that were separated chromatographically at 3.9 and 5.0 m, matched the theoretical $m/z$ ([M-H]: 159.029 $m/z$) predicted for mono-methyl-KG, and were only detected in DMKG-treated cells (Supplementary Fig. 2d). Analysis of tandem MS fragment spectra (MS2) provided evidence that each peak represented a mono-ester form of KG and represented either 1-methyl-KG (1MKG) or 5-methyl-KG (5MKG) (Supplementary Fig. 2e, f). 5MKG, generated by hydrolysis of the ester group proximal to the α-ketone, was ~16-fold more abundant than 1MKG (Fig. 2c). This observation supports the broader conclusion that α-ketoesters are more labile. Indeed, Zengeya et al. showed that regiospecific 1-TFMB-KG esters were significantly more

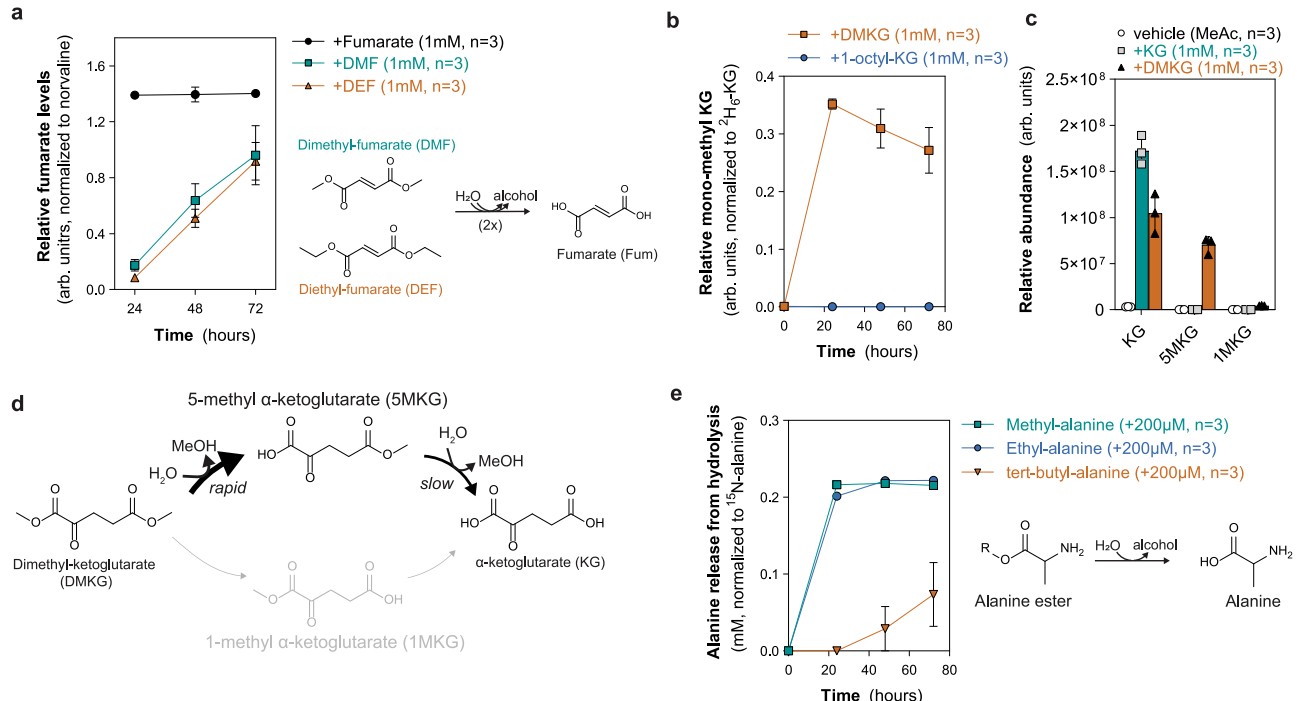

**Fig. 2 Ester hydrolysis is common and dependent on hydrophobicity of ester group. a** Time course of relative fumarate levels resulting from double hydrolysis of 1 mM of dimethyl-fumarate (DMF) or diethyl-fumarate (DEF) over 72 h compared to 1 mM of fumarate. Hydrolysis was performed in cell-free conditions. Data are mean ± s.d., $n = 3$ biologically independent experiments. **b** Time course of relative (1/5)-methyl KG levels quantified by GC-MS after cell-free hydrolysis of 1 mM of DMKG or 1-octyl-KG over 72 h. Data are mean ± s.d., $n = 3$ biologically independent experiments. **c** Relative levels of KG, 5-methyl-KG (5MKG), and 1-methyl-KG (1MKG) in 8988T cells treated with methyl acetate or 1 mM of KG or DMKG for 3 h quantified by LC-MS. Data are mean ± s.d., $n = 3$ biologically independent samples. **d** Schematic depicting rapid hydrolysis of the ester moiety proximal to the α-ketone of KG, producing 5-methyl-α-ketoglutaric acid (5MKG). Subsequent hydrolysis of the distal methyl-ester occurs at a slower rate and releases α-ketoglutaric acid (KG).
**e** Absolute alanine concentration released after cell-free hydrolysis of 0.2 mM methyl-alanine, ethyl-alanine, or tert-butyl-alanine over 72 h. Tert-butylated alanine exhibits a slower hydrolysis rate. Data are mean ± s.d., $n = 3$ biologically independent experiments.

susceptible to non-enzymatic hydrolysis relative to 5-TFMB-KG esters, owing to the increased acidity and leaving group potential of the 1-carboxylate of KG[30]. We also observed a significant accumulation of intracellular KG in both KG- and DMKG-treated cells after 3 h (Fig. 2c). However, DMKG treatment resulted in significantly less intracellular KG relative to KG treatment, because DMKG hydrolyzed to relatively equal amounts of KG and 5-methyl-KG after 3 h (Fig. 2c). Notably, a peak corresponding to the theoretical $m/z$ ([M-H]: 173.045 $m/z$) of DMKG was not detected above baseline in DMKG-treated cells, suggesting that DMKG may completely hydrolyze to KG or mono-esterified KG within this time frame. Taken together, our data suggest that hydrolysis of DMKG occurs immediately upon exposure to aqueous conditions and exists intracellularly as either KG or 5-methyl-KG (Fig. 2d).

To explore whether spontaneous hydrolysis was common to structurally distinct esters, we conducted cell-free hydrolysis experiments using esterified alanine analogs. Similar to our previous results, we observed hydrolysis of all alanine esters tested (Fig. 2e). However, compared to methyl- and ethyl-esterified alanine, which hydrolyzed rapidly and completely to alanine after 24 h, tert-butyl-alanine displayed significant stability in aqueous conditions suggesting that steric hindrance offered by large, aliphatic ester groups confers protection from hydrolysis (Fig. 2e). A major concern with using metabolite esters is potential toxicities associated with the released alcohol and/or aldehydes produced during their metabolism (e.g., formaldehyde from methanol). Indeed, diroximel fumarate, a heterotypic di-esterified fumarate, was recently approved by the FDA for relapsed-remitting MS in 2019 and consists of a methyl and

2-hydroxyethyl succinimide modification, the latter of which reduces methanol release during hydrolysis to MMF and is thought to lessen gastrointestinal side effects[40]. However, we observed no negative proliferative impacts in MiaPaCa2 cells when treated with either 2 or 10 mM of methyl acetate or tert-butyl acetate that would produce methanol or tert-butanol upon hydrolysis, respectively (Supplementary Fig. 2g). Collectively, our results demonstrate that spontaneous hydrolysis of esterified substrates is common and hydrolytic rates must be empirically determined for each metabolite, esterified analog, and environmental condition.

**KG esters impact metabolism independent of KG.** KG is an important intermediate of the TCA cycle in mitochondria and acts as the major de-aminated substrate involved in transaminase reactions linking amino acid anabolism and catabolism[41]. In addition, KG is an obligatory co-substrate for a large oxygen-dependent family of enzymes broadly categorized as KG-dependent dioxygenases that catalyze the hydroxylation of DNA, proteins, and lipids[42]. We hypothesized that KG and its analogs may drive an increase in cellular respiration due to uptake and metabolism by oxygen-dependent pathways, including the electron transport chain (ETC) and/or KG-dependent dioxygenases. To explore this, we conducted respirometry analysis in 8988T cells acutely stimulated with either vehicle (methyl acetate) or 1 mM of KG, DMKG, or 1-octyl-KG for ~60 m once baseline oxygen consumption (OCR) and extracellular acidification (ECAR) rates were established. Neither respiration nor glycolytic metabolism was significantly altered following KG injection, suggesting that imported KG may not stimulate flux

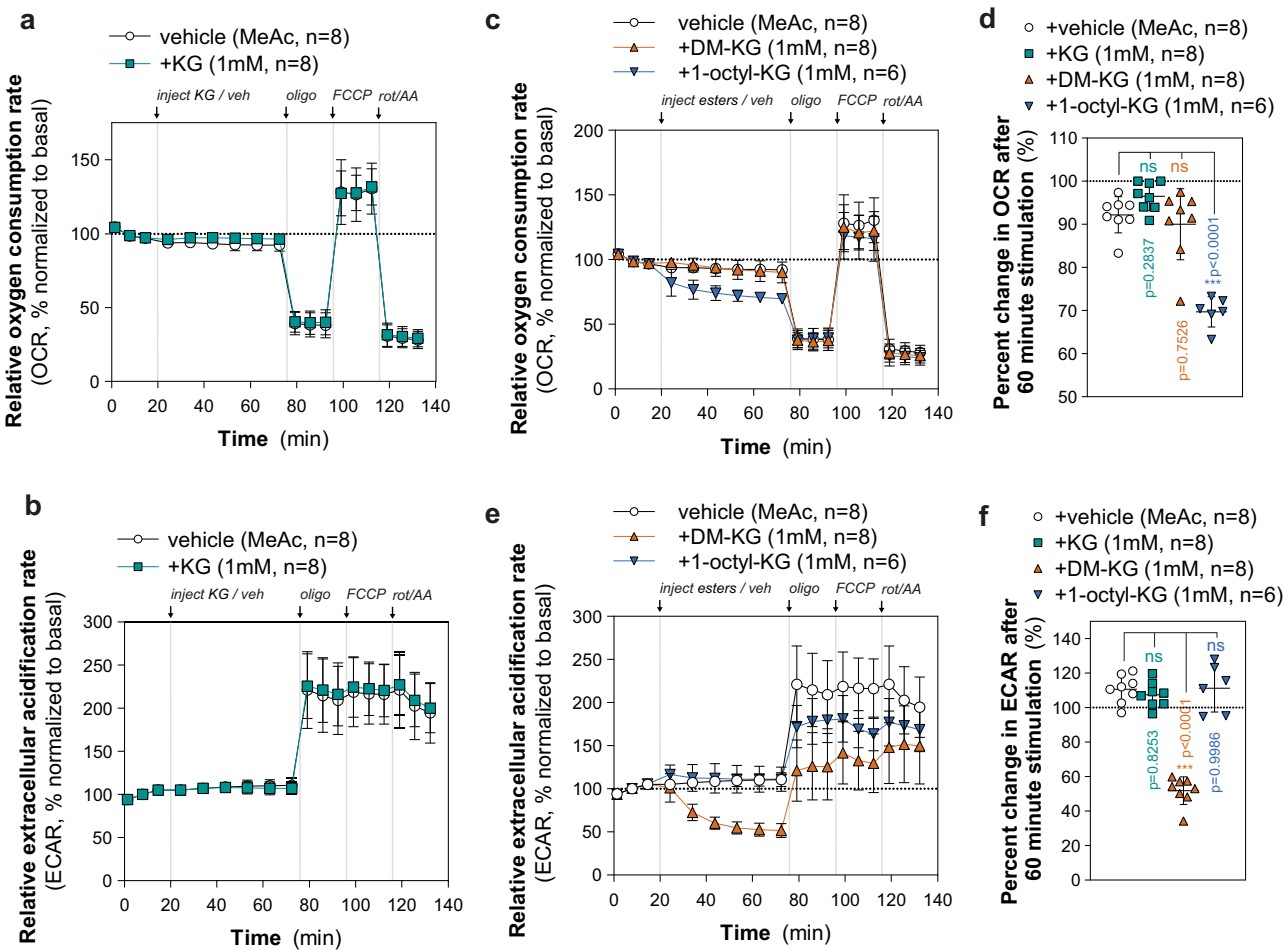

**Fig. 3 Esterified KG analogs exhibit KG-independent effects on glycolytic and mitochondrial metabolism. a, b** Relative oxygen consumption rate (OCR, **a**) and extracellular acidification rate (ECAR, **b**) of 8988T cells upon stimulation with 1 mM of KG or methyl acetate after the establishment of baseline OCR for ~20 m (basal). Following 60-m stimulation, sequential injections of oligomycin (oligo), FCCP, and rotenone with antimycin A (rot/AA) were made. Data are mean ± s.d. of $n = 8$ biologically independent experiments. Data were normalized to the average of the three basal OCR and ECAR values from each experiment. **c** Relative OCR of 8988T cells upon stimulation with methyl acetate or 1 mM of DMKG or 1-octyl-KG. Data are mean ± s.d. of $n = 6$ (1-octyl-KG) or $n = 8$ (methyl acetate, DMKG) biologically independent experiments. **d** Percent change in OCR of 8988T cells stimulated with methyl acetate or 1 mM of KG, DMKG, or 1-octyl-KG for 60 m. Data are plotted as a percent relative to baseline OCR for each biologically independent experiment prior to stimulation. Data are mean ± s.d. of $n = 6$ (1-octyl-KG) or $n = 8$ (methyl acetate, DMKG) biologically independent experiments; significance determined by one-way ANOVA using Dunnett's multiple comparisons test, n.s. $p > 0.05$, ***$p < 0.0001$. **e** Relative ECAR of 8988T cells upon stimulation with methyl acetate or 1 mM of DMKG or 1-octyl-KG. Data are mean ± s.d. of $n = 6$ (1-octyl-KG) or $n = 8$ (methyl acetate, DMKG) biologically independent experiments. **f** Percent change in ECAR of 8988T cells stimulated with methyl acetate or 1 mM of KG, DMKG, or 1-octyl-KG for 60 m. Data are plotted as a percent relative to baseline ECAR for each biologically independent experiment prior to stimulation. Data are mean ± s.d. of $n = 6$ (1-octyl-KG) or $n = 8$ (methyl acetate, DMKG) biologically independent experiments; significance determined by one-way ANOVA using Dunnett's multiple comparisons test, n.s. $p > 0.05$, ***$p < 0.0001$.

through KG-dependent pathways (Fig. 3a, b). However, esterified KG exhibited significant analog-specific impacts on cellular metabolism upon acute stimulation. Injection of 1-octyl-KG led to an acute and significant decrease in oxygen consumption by ~30% that was not observed in cells treated with methyl acetate, KG, or DMKG (Fig. 3c, d). Furthermore, injection of DMKG significantly inhibited glycolytic flux by ~50% after 60 m of stimulation and significantly inhibited oligomycin-induced increases in glycolysis, suggesting that DMKG impairs the glycolytic potential of cells (Fig. 3e, f; Supplementary Fig. 3a). Notably, these distinct metabolic impairments caused by 1-octyl-KG and DMKG were most likely independent of released KG from hydrolysis of the ester analogs as the addition of KG did not illicit either effect. To determine whether either analog cause cellular toxicity, we cultured 8988T cells for five days with methyl acetate or 1 mM KG, DMKG, or 1-octyl-KG. No significant effect on proliferation

was seen for KG or DMKG, but 1-octyl-KG caused a significant and near-complete suppression of proliferation (Supplementary Fig. 3b). Cellular toxicity and decreased cellular respiration caused by 1-octyl-KG, and not DMKG nor TFMB-KG, has also been reported in U2OS human osteosarcoma cells[10]. These results highlight potential "off-target" effects of KG esters on cellular metabolism that may influence phenotypes associated with their use.

**α-ketone esters cause extracellular acidification.** We hypothesized that the rapid and spontaneous hydrolysis of esterified KG analogs may indirectly affect cellular metabolism, either through the release of alcohol or another mechanism. Synthesis of ester analogs is typically achieved by an acid-catalysis mechanism using chlorotrimethylsilane, a metabolite, and alcohol[43–45]. The recent development of modular synthesis strategies enable

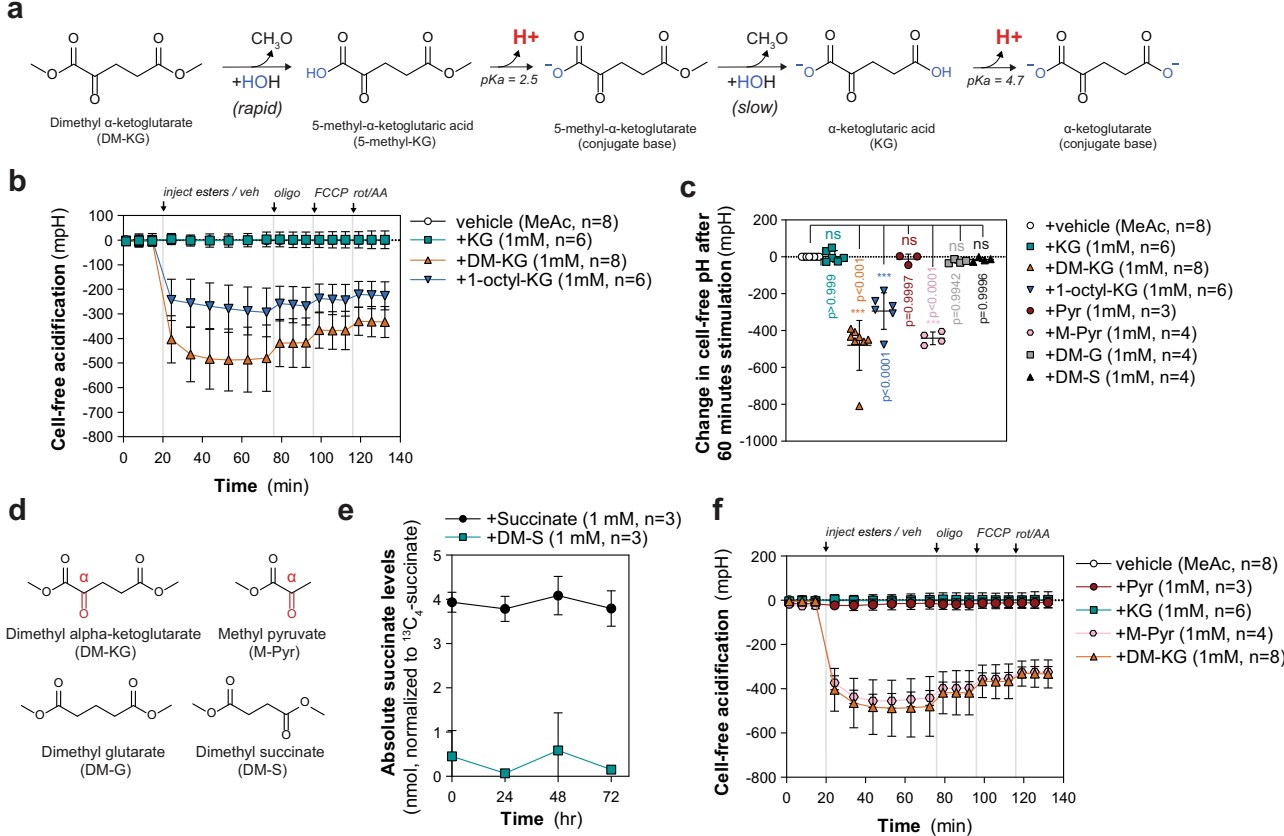

**Fig. 4 α-ketoesters significantly acidify aqueous media as a result of rapid hydrolysis and de-protonation. a** Schematic depicting the mechanism of DMKG hydrolysis, formation of α-ketoacid, and subsequent de-protonation that results in significant acidification of aqueous media. The final product upon complete hydrolysis is the conjugate base of α-ketoglutarate (KG). **b** Acidification of media without cells after injection of methyl acetate or 1 mM of KG, DMKG, or 1-octyl-KG in units of mpH. Data are mean ± s.d. of $n = 6$ (KG, 1-octyl-KG) or 8 (methyl acetate, DMKG) biologically independent experiments. **c** Change in media pH (in mpH) after 60-m stimulation with methyl acetate or 1 mM of KG, DMKG, 1-octyl-KG, pyruvate (Pyr), methyl pyruvate (M-Pyr), dimethyl-glutarate (DM-G), or dimethyl-succinate (DM-S). Data are mean ± s.d. of $n = 3$ (Pyr), 4 (M-Pyr, DM-G, DM-S), 6 (KG, 1-octyl-KG), or 8 (methyl acetate, DMKG) biologically independent experiments; significance determined by one-way ANOVA using Dunnett's multiple comparisons test, n.s. $p > 0.05$, ***$p < 0.0001$. **d** Chemical structures of DMKG, methyl pyruvate (M-Pyr), dimethyl-glutarate (DM-G), and dimethyl-succinate (DM-S). The α-keto group is colored in red and marked with "α" symbol for emphasis. **e** Time course of absolute succinate levels resulting from double hydrolysis of 1 mM of dimethyl-succinate (DM-S) over 72 h compared to 1 mM of succinate. Hydrolysis was performed in cell-free conditions. Data are mean ± s.d., $n = 3$ biologically independent experiments. **f** Acidification of media without cells after injection of methyl acetate or 1 mM of Pyr, M-Pyr, KG, or DMKG in units of mpH. Data from Fig. 4a included for KG and DMKG for comparison to acidification caused by M-Pyr. Data are mean ± s.d. of $n = 3$ (Pyr), 4 (M-Pyr), 6 (KG), or 8 (methyl acetate, DMKG) biologically independent experiments. **b**, **f** Cell-free assays run in parallel with in-cell assays from Fig. 3, which is why sequential injections of oligomycin, FCCP, and rotenone with antimycin A were included. Acidification assays conducted in DMEM containing HEPES (5 mM) buffer. Change in mpH quantified for each biologically independent experiment by background subtracting vehicle from experimental data and averaging across all experiments.

production of asymmetric and/or regiospecific KG esters[11]. Hydrolysis of esters in aqueous environments can occur either through an acid- or base-catalyzed mechanism and is reported to involve autoionization of water into hydroxide and hydronium ions that participate in the hydrolysis reaction[46]. The product of either catalytic mechanism is the resulting acid (e.g., α-ketoglutaric acid), which at physiological pH may be expected to deprotonate to its anionic, conjugate base form depending on the pKa of the carboxylic acid(s) (Fig. 4a). Thus, we hypothesized that the rapid nature of hydrolysis of DMKG to α-ketoglutaric acid with a ~pKa of 3.87 (C1 pKa ~ 2.5; C5 pKa ~ 4.7) will contribute to a significant increase in acidity[47–49]. In contrast, the disodium salt of α-ketoglutaric acid is in its anionic, conjugate base form and would not have a similar effect (Supplementary Fig. 4a). To better understand the hydrolysis mechanism and potential KG-independent effects that DMKG or 1-octyl-KG may have on cellular metabolism, we monitored pH dynamically

following acute injection in cell-free conditions in the presence of a HEPES buffer, similar to previous OCR and ECAR measurement conditions. Injection of either DMKG or 1-octyl-KG significantly acidified culture media by ~500 and ~300 mpH, respectively (Fig. 4b). Injection of KG did not illicit a similar effect on media pH, suggesting that proton release from α-ketoglutaric acid produced during hydrolysis of KG analogs is likely the source of extracellular acidification (Fig. 4b, c).

We sought to explore the nature of proton release by investigating the hydrolysis of other, structurally similar methyl esters. The ketone group of α-ketoesters is more electrophilic owing to the electron-withdrawing properties of the ester group, making esterified KG potentially more susceptible to nucleophilic attack and exhibit faster hydrolysis kinetics[30,50]. To determine if the α-ketone carboxylic acid contributes to significant proton release, we conducted cell-free acidification experiments using structurally similar dicarboxylates that lack an α-keto group,

including dimethyl-glutarate (DM-G) and dimethyl-succinate (DM-S) (Fig. 4d). The pKa of succinic acid (C1 pKa ~ 4.2; C4 pKa ~ 5.6) and glutaric acid (C1 pKa ~ 4.3; C5 pKa ~ 5.4) are well below physiologic pH and expected to exist mainly in the deprotonated form similar to KG[49]. However, injection of either DM-G or DM-S did not lead to significant acidification of media over 60 m relative to vehicle (Fig. 4c; Supplementary Fig. 4b). Unlike DMKG, we did not observe significant cell-free hydrolysis of DM-S to succinic acid after 72 h by GC-MS, suggesting that hydrolysis and subsequent deprotonation may be too slow to exceed the media buffering capacity over this time frame (Fig. 4e). To determine if proton release was common to other α-ketoesters, we performed cell-free acidification experiments using methyl-pyruvate (M-Pyr) or sodium pyruvate (conjugate base) (Fig. 4d). Injection of 1 mM methyl pyruvate led to significant acidification by ~500 mpH, whereas injection of sodium pyruvate did not (Fig. 4f). Taken together, our data suggest that α-ketone containing metabolite esters are vulnerable to rapid spontaneous hydrolysis and lead to significant proton release in aqueous environments. To determine if methyl pyruvate also inhibits glycolytic metabolism, we performed respirometry analysis in 8988T cells stimulated with either pyruvate or methyl pyruvate. Similar to DMKG, injection of methyl pyruvate significantly inhibited the glycolytic flux of 8988T cells (Supplementary Fig. 4c). However, in contrast to KG, pyruvate led to significant impacts on mitochondrial and glycolytic flux (Supplementary Fig. 4c, d), consistent with pyruvate acting as a major electron acceptor in central carbon metabolism, convoluting effects specific to the methyl-ester[51].

**Utilization of imported KG by cytosolic dioxygenases**. To better understand potential off-target metabolic effects of DMKG relative to KG, we conducted targeted metabolomics by LC-MS of 8988T cells treated with methyl acetate or 1 mM of either KG or DMKG for 3 h. Only two metabolites, KG and succinate, were significantly increased in KG-treated cells; however, DMKG treatment resulted in extensive metabolomic alterations indicating that the activity of these two compounds is distinct (Fig. 5a, b). DMKG treatment led to a significant increase in mitochondrial metabolites, including succinate, fumarate, and malate, as well as aspartate which suggests that DMKG may impact mitochondrial TCA cycle metabolism (Fig. 5b). Furthermore, we observed significant increases in several α-ketone containing metabolites, including pyruvate and oxoadipic acid, and propionylcarnitine, which accumulates as a result of propionyl-CoA production (Fig. 5b). Notably, these three intermediates represent endogenous α-ketones with varying carbon lengths, suggesting that DMKG, or 5-methyl-KG, may affect the activity of dehydrogenases involved in the metabolism of α-ketoacids by virtue of structural homology to endogenous substrates. Among the other significantly altered metabolites, cystine and L-acetylcarnitine were significantly depleted following acute DMKG treatment (Fig. 5b). Thiol-containing metabolites, such as cyst(e)ine and glutathione, can act as potent nucleophiles and may interact with electrophilic compounds such as esterified KG. Furthermore, L-acetylcarnitine can serve as a pool of cellular acetate, and acylcarnitines can directly supply mitochondria with anaplerotic substrates[52]. Taken together, esterified KG derivatives exhibit significant spurious effects on metabolism independently of KG itself.

Given the importance of KG for intracellular metabolism, we sought to better understand how imported KG is utilized by cells. In agreement with our previous results, intracellular KG levels were significantly increased upon the addition of KG; however, succinate levels also increased indicative of KG utilization by cells

(Fig. 5a). Succinate is the decarboxylated product of KG-dependent reactions, including the mitochondrial TCA cycle enzyme α-ketoglutarate dehydrogenase (KGDH) and all KG-dependent dioxygenases (Fig. 5c). In addition, KG can spontaneously decarboxylate in the presence of intracellular oxidants, including hydrogen peroxide; however, cytosolic levels of $H_2O_2$ are low (0.1–0.5 μM) under homeostatic conditions and, thus, not expected to contribute significantly to succinate production[53,54]. We analyzed our stable-isotope dilution data and found a significant incorporation of imported KG into cellular KG and succinate pools (Fig. 5d; Supplementary Fig. 5a). To better understand how KGDH, $H_2O_2$, and/or KG-dependent dioxygenases were contributing to KG utilization, we conducted stable-isotope dilution in 8988T cells cultured with $^{13}C_5$-glutamine in anoxic culture conditions, as oxygen is a required co-substrate for KG metabolism and endogenous $H_2O_2$ production (Fig. 5c). Anoxic culture conditions were achieved by depletion of oxygen using a palladium catalyst to maintain oxygen levels at 3–5 ppm (<0.001% $O_2$) in a sealed glove box, as previously[55]. In anoxia, KG decarboxylation was significantly inhibited suggesting that KG utilization is primarily through oxygen-dependent reaction(s) (Fig. 5e). To determine if imported KG or DMKG could act as an anaplerotic source for mitochondrial TCA cycle metabolism, we quantified $^{13}C$ label-dilution into downstream TCA cycle intermediates (e.g., fumarate, malate, glutamate, and citrate) in HuPT4, 8988S, 8988T, and AsPC1 cell lines as well as RAW264.7 macrophages, given the broad usage of KG analogs in immune cell types (Fig. 5f). While we observed significant incorporation of KG into succinate pools across all cell lines except for RAW264.7 macrophages, we quantified minimal contribution into TCA cycle metabolites (0.5–2%) in all KG-supplemented cells (Fig. 5g, h; Supplementary Fig. 5b). To determine if this was a cancer cell line phenomenon, we repeated the stable-isotope dilution in a non-transformed mouse embryonic fibroblast (MEF) cell line, and we observed similar import of KG and decarboxylation to succinate but minimal contribution to TCA cycle metabolites (~2–3%) (Supplementary Fig. 5c)[56]. These data suggest that KG import is context-dependent and does not significantly contribute to mitochondrial metabolism in normal culture conditions. Thus, the succinate derived from imported KG is likely independent of KGDH and is extra-mitochondrial.

To gain a deeper understanding of the spurious metabolic effects of DMKG, we conducted an untargeted feature detection in our LC-MS data for 8988T cells treated with 1 mM of DMKG compared to KG-treated cells. A significant number of altered features were detected and unique to DMKG-treated cells (Supplementary Fig. 5d). Because DMKG transiently exists as intracellular 5-methyl-KG and KG (Fig. 2c), we hypothesized that methylated KG may retain passive diffusion properties and contribute to mitochondrial metabolism directly. We manually searched for predicted features corresponding to the methylated TCA intermediates (e.g., methyl-succinate), which we hypothesized could arise from the enzymatic metabolism of 5-methyl-KG. Strikingly, we found two features with parent ions that matched the predicted *m/z* for methyl-succinate (131.034 *m/z*, negative mode) and methyl-fumarate (129.018 *m/z*, negative mode) that were not detected in either KG or vehicle conditions (Supplementary Fig. 5e, f). To confirm the identity of these untargeted features, we analyzed the MS2 fragmentation spectra and identified neutral losses that were consistent with the structures of methyl-succinate and methyl-fumarate (Supplementary Fig. 5g, h). These data indicate that 5-methyl-KG can likely be directly metabolized within cells to form methylated TCA intermediates (Supplementary Fig. 5f); however, these features must be validated and quantified with chemical standards to determine the amount produced by DMKG treatment. These methylated intermediates may further explain the

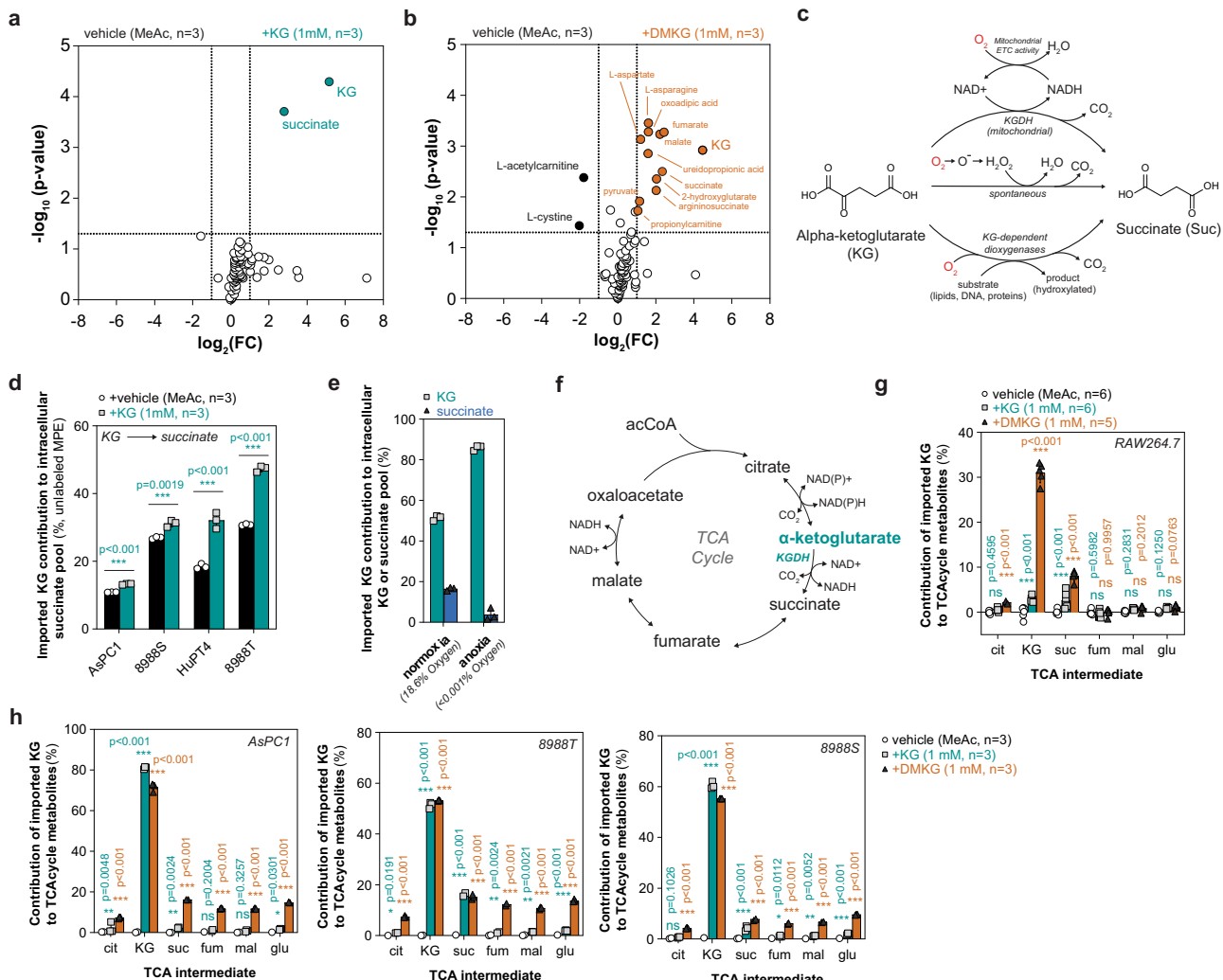

**Fig. 5 KG utilization by cells and spurious metabolic effects of DMKG. a**, **b** Volcano plot of metabolomic changes in 8988T cells treated with methyl acetate or 1 mM of KG or DMKG for 3 h. **a** Only KG and succinate were significantly increased in KG-treated cells, whereas 14 metabolite levels changed significantly in response to DMKG treatment in (**b**). Metabolite changes were considered significant if $\log_2$ fold change (FC) was $\geq 1$ or $\leq -1$ and adjusted $p$ value < 0.05; significance was determined using the Wald test (DESeq2) corrected for multiple comparisons. **c** Schematic depicting possible metabolic routes of KG decarboxylation to succinate (Suc). Mitochondrial KG dehydrogenase (KGDH) catalyzes the NAD+-dependent oxidative decarboxylation, producing NADH which fuels ETC activity. KG can oxidatively decarboxylate to Suc in the presence of cellular hydrogen peroxide, which can be produced from superoxide free radicals. KG can participate in hydroxylation reactions catalyzed by the large family of KG-dependent dioxygenases, which decarboxylate KG to Suc. **d** Contribution of imported KG to succinate pools quantified by measuring %(100-MPE) of succinate labeling from $^{13}C_5$-glutamine in AsPC1, 8988S, HuPT4, and 8988T cells cultured for 24 h. Data are mean ± s.d., $n = 3$ biologically independent samples; significance determined by two-tailed Student's $t$-test corrected for multiple comparisons, ***$p$ < 0.0001. **e** Contribution of imported KG to intracellular KG and succinate pools in 8988T cells cultured in normoxic (18.6% oxygen) or anoxic (<0.001% oxygen) conditions for 24 h with $^{13}C_5$-glutamine with or without 1 mM KG; normoxic data plotted from Fig. 5h. Data are plotted as the difference in %MPE from $^{13}C_5$-glutamine between KG-supplemented and control (MeAc) conditions, mean ± s.d., $n = 3$ biologically independent samples. **f** Schematic depicting reactions of the mitochondrial TCA cycle including KGDH; acetyl-CoA (acCoA). **g**, **h** Contribution of imported KG or DMKG to mitochondrial TCA cycle intermediates, including succinate (suc), fumarate (fum), malate (mal), glutamate (glu), and citrate (cit). Percent contribution calculated by measuring the difference in %MPE from $^{13}C_5$-glutamine between KG- or DMKG- supplemented and control (MeAc) conditions in RAW264.7 macrophages (**g**) or AsPC1, 8988T, and 8988S pancreatic cancer cells (**h**) cultured for 24 h with or without 1 mM unlabeled KG. Data are mean ± s.d., $n = 3$ biologically independent samples; significant determined by two-way ANOVA using Dunnett's multiple comparisons test, n.s. $p$ > 0.05, *$p$ < 0.05, **$p$ < 0.01, ***$p$ < 0.001.

incongruency between the metabolic effects of DMKG versus KG treatment that we observed. As further support for the mitochondrial permeability of 5MKG/DMKG, we found that DMKG contributed to TCA metabolites beyond succinate in many of the cell lines examined, including citrate, fumarate, malate, and glutamate (Fig. 5g, h). Notably, RAW264.7 macrophages did not share this phenotype, and DMKG contribution was limited to succinate in this context (Fig. 5g). Analysis of oxidative (M + 4) and

reductive (M + 5) citrate isotopologue enrichment suggests that DMKG treatment most notably contributes to oxidative TCA cycle metabolism (e.g., dilution of M + 4 citrate) (Supplementary Fig. 5i, j). Further, intracellular levels of malate, fumarate, and aspartate—which are produced by oxidative mitochondrial metabolism—were significantly and mostly elevated in DMKG-treated cells (Supplementary Fig. 5j, k). Taken together, these results highlight that DMKG contributes to cytosolic and mitochondrial metabolism

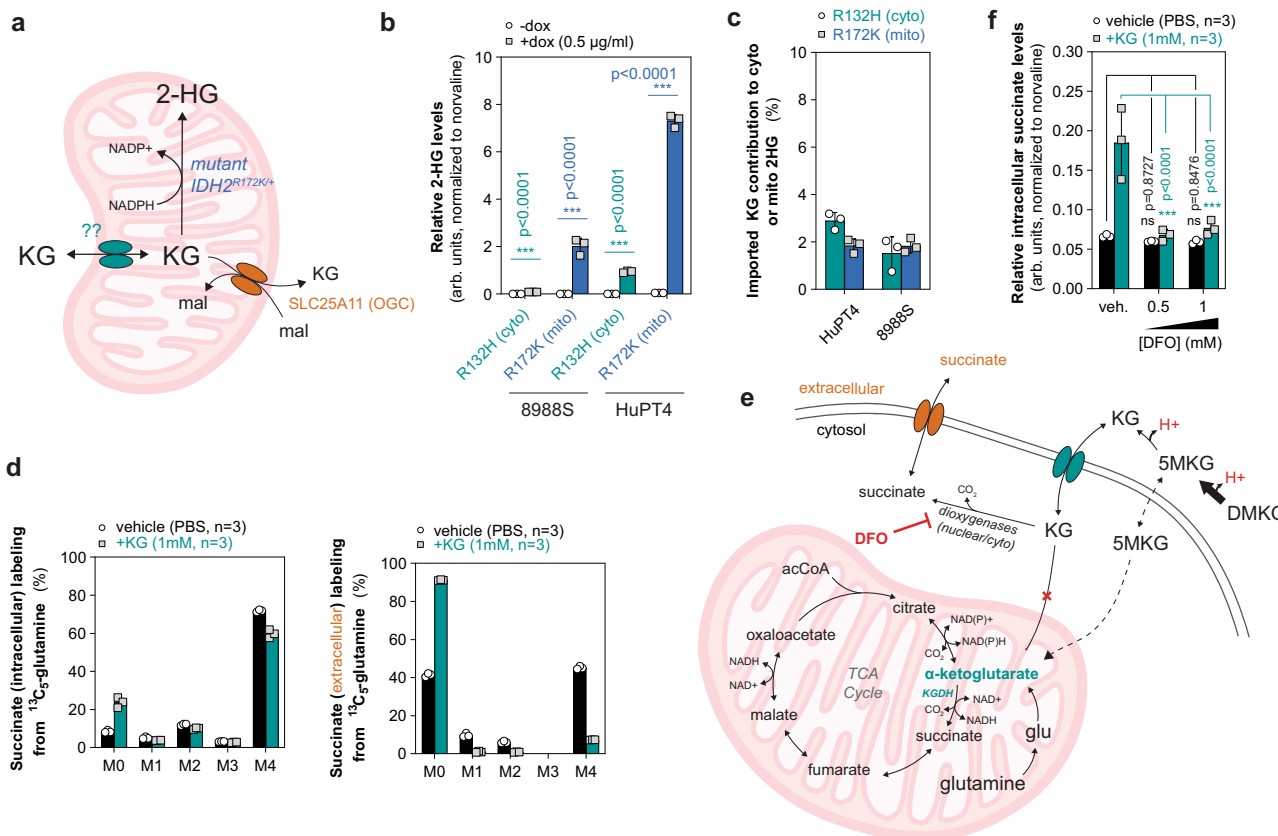

**Fig. 6 KG metabolism is functionally compartmentalized in cells. a** Schematic of mitochondrial production of 2-hydroxyglutarate (2-HG) from mutant IDH2$^{R172K/+}$ as a sensor of mitochondrial KG labeling. SLC25A11/OGC is an integral component of the malate-aspartate shuttle and exchanges mitochondrial KG with cytosolic malate; however, whether mitochondrial KG can import into mitochondria is unknown. **b** Relative cytosolic or mitochondrial 2-HG production from dox-inducible expression of IDH1-R132 (R132H, cyto) or IDH2-R172K (R172K, mito), respectively, in HuPT4 and 8988S cells after 24 h dox-induction (0.5 µg/ml) normalized to norvaline levels. Data are mean ± s.d., $n = 3$ biologically independent samples; significant determined by two-tailed Student's $t$-test corrected for multiple comparisons, ***$p < 0.0001$. **c** Minimal label incorporation from imported KG into cytosolic or mitochondrial 2-HG production from R132H or R172K labeled with $^{13}C_5$-glutamine with or without 1 mM of KG for 24 h in 8988S and HuPT4 cells. IDH1-R132H or IDH2-R172K expression was induced using 0.5 µg/ml at the time of labeling. Data are plotted as the difference in %(100-MPE) from $^{13}C_5$-glutamine between KG-supplemented and control (PBS) conditions, which represents the percentage of carbon that KG contributes to 2-HG production, mean ± s.d., $n = 3$ biologically independent samples. **d** Mass isotopologue distribution of intracellular (left panel) or extracellular (right panel) succinate labeling from HuPT4 cells cultured with $^{13}C_5$-glutamine with or without 1 mM of unlabeled KG for 24 h. Secreted succinate is a surrogate measurement of the cytosolic pool. Data are mean ± s.d., $n = 3$ biologically independent samples. **e** Schematic of DMKG hydrolysis and deprotonation and cellular import of KG and 5MKG and metabolism by cytosolic and/or nuclear dioxygenases to produce succinate, which can be secreted. Minimal contribution of imported KG was observed in mitochondrial TCA cycle intermediates, but DMKG/5MKG was able to contribute to the TCA cycle. Deferoxamine (DFO) is an iron chelator, which broadly inhibits KG-dependent dioxygenase activity. Acetyl-CoA (acCoA), glutamate (glu). **f** Treatment of 8988T cells with DFO (500 µM or 1 mM) for 24 h significantly inhibits the increase in intracellular succinate levels from KG supplementation. Data are representative mean ± s.d., $n = 3$ biologically independent samples from two independent experiments; significance determined by two-way ANOVA using Dunnett's multiple comparisons test, n.s. $p > 0.05$, ***$p < 0.0001$.

and/or dioxygenase activity, whereas extracellular KG uptake and metabolism may specifically fuel cytosolic pathways.

**Functional compartmentalization of KG metabolism.** To quantify the compartment-specific KG metabolism, we used a previously described reporter system capable of differentiating cytosolic and mitochondrial labeling of 2-hydroxyglutarate (2-HG), the product of IDH1$^{R132H/+}$ (cytosolic) and IDH2$^{R172K/+}$ (mitochondrial) mutant enzymes[57]. While this reporter was used in the past as a readout of compartmentalized $^2$H-labeled NADPH by surrogate measurements of 2-HG, we hypothesized that this system could also be applied to differentiate cytosolic and mitochondrial KG pools, as KG is the carbon backbone of 2-HG (Fig. 6a). We ectopically expressed dox-inducible IDH1-R132H or

IDH2-R172K in 8988S and HuPT4 cell lines. All cell lines produced a detectable level of 2-HG over baseline after 24 h dox-induction of mutant IDH1-R132H or IDH2-R172K (Fig. 6b). We performed stable-isotope dilution experiments using $^{13}C_5$-glutamine and 1 mM of KG for 24 h with doxycycline and observed significant incorporation of KG into intracellular KG pools consistent with our previous results (Supplementary Fig. 6a). Strikingly, very minimal (1–3%) dilution of 2-HG labeling from imported KG was observed in both cytosolic and mitochondrial produced 2-HG (Fig. 6c; Supplementary Fig. 6b, c). In addition, KG supplementation did not significantly affect 2-HG induction (Supplementary Fig. 6d). These data confirm that KG transport into mitochondria from the cytosol is limited, and that the known mitochondrial KG antiporter (SLC25A11) predominantly operates as a KG exporter in normal culture conditions (Fig. 6a). In

addition, our data suggest that KG for IDH1 activity may be subject to substrate channeling as imported, presumably cytosolic KG, did not significantly contribute to mutant IDH1 production of 2-HG (Fig. 6c). To determine if a separate cytosolic pool of KG and succinate exists, we measured label incorporation from KG into exported succinate in HuPT4 cells. While the intracellular pool of succinate was significantly labeled by glutamine and KG import contributed ~15%, secreted succinate was almost entirely labeled by imported KG (~92%) (Fig. 6d). These data suggest that the cellular pool of KG is not well-mixed, and functionally distinct cytosolic/nuclear and mitochondrial pools exist (Fig. 6e).

Furthermore, our data provide evidence that some KG-dependent enzymes may acquire KG through substrate channeling likely from functional interaction with upstream enzymes, including cytosolic transaminases (e.g., GOT1, PSAT1) and/or GLUD1. To functionally determine whether cytosolic and/or nuclear dioxygenases contribute to the utilization of imported KG, we treated 8988T cells with 1 mM of KG and deferoxamine (DFO) to chelate iron required for KG-dependent dioxygenase activity[58]. In response to DFO in non-KG-supplemented conditions, we observed no significant difference in succinate levels (Fig. 6f). However, the increase in intracellular succinate levels in KG-supplemented cells was significantly suppressed by DFO treatment, confirming that utilization of imported KG was mainly through cytosolic and/or nuclear dioxygenase activity (Fig. 6e, f).

## Discussion

Our work demonstrates that many metabolite esters, including KG analogs, are vulnerable to rapid, spontaneous hydrolysis in aqueous conditions and are likely taken up by cells in metabolite form or transiently as monoesters (e.g., mono-methyl-KG). While it is often assumed that cell membranes are impermeable to KG, our data suggest that many cell lines can take up KG and likely use specific transporter(s) to facilitate its diffusion across cell membranes. Esterified KG analogs accumulated in cell lines that could take up endogenous KG and those that did not (e.g., RAW264.7 macrophages), suggesting that uptake involves KG transporter(s) and/or passive diffusion across the plasma membrane. Transport of KG and other dicarboxylates (e.g., succinate) is best described in the kidney and is mediated by the sodium-dependent dicarboxylate family of transporters, including NaDC-1/SLC13A2 and NaDC-3/SLC13A3 expressed in luminal and basolateral membranes of proximal tubule cells, respectively[59–61]. Notably, expression of NaDC-1 was not observed in renal carcinomas, suggesting that dicarboxylate transport may be silenced during tumorigenesis or may involve different transporter(s)[60]. Other transport systems that utilize KG as a co-substrate are also reported. For example, SLC25A11/OGC facilitates mitochondrial KG antiport with malate as an integral component of the malate-aspartate shuttle[62]. Furthermore, SLC22A6/OAT1 and SLC22A8/OAT3 utilize KG as an exchange factor for importing a wide range of substrates, including biogenic amines and several drugs, and are localized predominantly in the kidney and choroid plexus[63]. Not all of the cell types evaluated in our study were able to take up KG in its endogenous form. For example, supplemented KG did not significantly contribute to intracellular KG pools in RAW264.7 macrophages (Fig. 5g), suggesting that these cells lack expression of a plasma membrane KG transporter. Previous studies by MacKenzie et al.[11], Tennant et al.[28], and Zengeya et al.[30] show that HEK-293, HCT116, and HeLa cells do not significantly take up KG and require octyl- and/or TFMB-modified analogs to increase intracellular KG levels and cause biological effects. We also observed considerable heterogeneity in KG uptake across the cell lines tested (Fig. 1d), emphasizing the

need to carefully quantify uptake capacity and metabolic utilization using stable-isotope labeling in each context. How KG is transported by cancer, immune, and other cell types and which transporter(s) are involved is not well understood and warrants future study.

Our data suggest that extracellular KG acts primarily as a substrate for cytosolic KG-dependent reactions and contributes minimally to mitochondrial TCA cycle metabolism in the cell lines examined and under standard culture conditions. These findings suggest that extramitochondrial KG-dependent reactions (e.g., TET-mediated DNA demethylation, EGLN-mediated prolyl hydroxylation, JmjC-mediated histone demethylation) may derive KG from a functionally distinct pool supplied through uptake, cytosolic generation, and/or mitochondrial synthesis and export. Recent work has demonstrated that stimulation of cytosolic PSAT1-dependent de novo serine synthesis provides nuclear KG for the removal of repressive H3K27me3 marks that lead to activation of epidermal differentiation[64]. Furthermore, knockdown or inhibition of the cytosolic transaminase GOT1 promoted Th17 cell differentiation towards induced Treg (iTreg) through inhibition of FOXP3 promoter demethylation, although this mechanism was proposed to act through 2-hydroxyglutarate[29]. KG is also a substrate for lysyl- and prolyl- hydroxylases involved in collagen production and maturation. When activated, hepatic and pancreatic stellate cells exhibit a myofibroblastic phenotype and deposit a collagen-rich extracellular matrix during chronic inflammatory conditions (e.g., alcoholic liver disease, pancreatitis) and in tumors of pancreatic and hepatic origin[65,66]. It has also been shown that upregulation of cytosolic KG sources, including GOT1 and GLUD1, is a critical component of hepatic stellate cell activation[67,68].

While KG and esterified analogs both contributed to intracellular succinate pools, indicative of cytosolic/nuclear dioxygenase activity, DMKG also uniquely contributed significantly to mitochondrial TCA cycle metabolism. These results suggest that mitochondrial membranes are permeable to methylated KG, likely as the mono-methylated form given the speed at which hydrolysis of the proximal $\alpha$-ketoester occurred. This is further supported by the accumulation of methylated TCA intermediates in DMKG-treated cells, including methyl-succinate and methyl-fumarate, which are presumably generated by the metabolism of mono-methyl-KG by mitochondrial TCA cycle enzymes. Dietary sources of esterified TCA intermediates exist, including diethyl-malate from fermented products such as beer and dimethyl-succinate from peanuts and roasted filberts, and may have "exposomic" effects on (patho)physiology[69]. However, DMKG did not contribute to mitochondrial TCA cycle metabolites in RAW264.7 macrophages, which may suggest that specific transporter(s) are required for mitochondrial import of 5MKG/DMKG. This is similar to a recent report by Fets et al.[33], which found that monocarboxylate transporter 2 (MCT2) was required for the import of methyloxalylglycine (MOG), a mono-methyl ester derived from the hydrolysis of dimethyloxalylglycine (DMOG) that is commonly used to inhibit prolyl hydroxylase (PHD) activity and stabilize HIF1α. Together, these data provide mechanistic evidence for how DMKG may rescue mitochondrial metabolism defects that arise from glutamine withdrawal and glutaminase or transaminase inhibition in other published contexts[13–15,21,22,24,26,27,29]. However, it is important to understand whether this contribution is unique to esterified KG, or if endogenously produced and/or supplemented KG can import into the mitochondrial matrix when substrates like glutamine are limited, which requires further study. Further, it is not well understood how physiological sources of KG may influence metabolism and/or dioxygenase activity. Plasma and cerebrospinal fluid levels of KG in healthy children and adults are

~10 μM and ~5 μM, respectively, and elevated plasma KG levels (~20–60 μM) have been reported in diabetic or pyruvate carboxylase deficient children[70,71]. We use a super-physiological concentration of 1 mM in this study to represent the concentrations of DMKG and other analogs used in published studies, which typically range from 1–4 mM up to 32 mM. However, these high concentrations may enrich for negative characteristics associated with KG esters, including acidification and effects induced by monoesters and/or the released alcohol(s). Minimizing analog concentrations may be preferred to reduce these impacts and to better match physiological conditions. It is possible that esterified analogs may outperform endogenous KG uptake at lower concentrations and/or have distinct biological effects depending on the context and mechanism of KG uptake. Previous studies using TFMB-, octyl-, and/or dimethyl- KG analogs report opposing effects on HIF1α stabilization—via modulation of the activity of prolyl hydroxylases (PHDs)—in different cellular contexts, conditions, and KG analog concentrations[11,28,30,31]. Further, TFMB alcohol alone, released during hydrolysis of TFMB-KG analogs, has also been shown to stabilize HIF1α[30].

Proton-coupled lactate secretion through the SLC16A family of monocarboxylate transporters (MCTs) is an important component of aerobic glycolysis[72]. Because the MCTs are passive transporters, the accumulation of protons and/or lactate in the extracellular environment is expected to inhibit MCT activity, and consequently, glycolysis-induced extracellular acidification rate[73]. Although DMKG, 1-octyl-KG, and M-Pyr analogs caused significant acidification, only DMKG and M-Pyr treatments significantly inhibited ECAR (Fig. 3c, d, Supplementary Fig. 4d, e). A low pH threshold may be necessary to inhibit MCT and/or glycolytic activity, and/or the decrease in OCR specifically in 1-octyl-KG-treated cells may drive a compensatory increase in glycolysis, convoluting the cellular response. In contrast, the increase in OCR from pyruvate or M-Pyr supplementation likely contributes to the compensatory decrease in ECAR observed in both conditions (Supplementary Fig. 4d, e). However, the pH threshold and/or metabolic effects of KG, pyruvate, methanol, octanol, and/or mono-methyl-KG that contribute to glycolysis inhibition warrants future study. The acidification was relatively modest in cells treated with 1 mM of DMKG, 1-octyl-KG, or M-Pyr but will likely scale with increasing concentrations. For comparison, the extracellular $pH_e$ of tumors typically ranges from 6.5 to 6.9[74]. Furthermore, extracellular acidification as low as 6.0 can occur during inflammation, which leads to direct effects on the metabolism and differentiation state of immune cells recruited to these acidified microenvironments[75]. Thus, although we do not know if this magnitude of acidification is directly responsible for glycolysis inhibition in this context, the impacts that KG analogs have on glycolysis and respiration occur independently of KG.

Other metabolite analogs have reported cellular effects that differ from their endogenous precursor. DMF is an immunomodulatory pro-drug therapeutic used to treat autoimmune disorders[76–79]. The therapeutic activity of mono-methyl fumarate is suggested to act through reactivity of the internal alkene with cellular thiols, such as glutathione, that leads to activation of a NRF2-dependent antioxidant response[80–83]. However, fumarate itself does not illicit a similar response, highlighting the unique properties of esterified fumarate[80]. Itaconate, a derivative of cis-aconitate produced by cis-aconitate decarboxylase (CAD), is a well-established immuno-modulatory metabolite and endogenous succinate dehydrogenase (SDH) inhibitor[84–86]. Esterified derivatives of itaconate (e.g., dimethyl-itaconate, 4-octyl-itaconate, 4-monoethyl-itaconate) are commonly used because of presumed poor cellular delivery of endogenous itaconate. However, several studies have shown that unmodified itaconate can be taken up by

several cell types, including immune, cancer, brain, and adipocytes[34,85,87–89]. Furthermore, itaconate esters can contribute to divergent cellular phenotypes as a result of differing analogs, incubation timing, and cellular contexts[84,90–92]. and only recently have itaconate and its analogs been comparatively examined[34]. These paradoxical effects and our data provide compelling evidence that α-ketoesters should be used with caution as surrogates for their respective metabolites (e.g., pyruvate, KG), and phenotypes described using DMKG, methyl-pyruvate, or other α-ketoesters should be confirmed using non-esterified metabolites, ester-derived alcohol(s), and/or orthogonal approaches.

## Methods

**Cell culture and proliferation.** The cell lines AsPC1, 8988S, HuPT4, PANC1, 8988T, MiaPaCa2, DanG, and RAW264.7 were obtained from ATCC or the DMSZ. Wild-type MEFs were a gift from the DePinho Laboratory (The University of Texas MD Anderson Cancer Center) and were generated as previously described[56]. 293T cells were a gift from William Hahn (Dana-Farber Cancer Institute). The routine culture was performed in humidified incubators at 37 °C and 5% $CO_2$ in DMEM (Corning) media supplemented with 10% fetal bovine serum (Atlanta Biologicals S11550H, Lot No. C18030) and 1% Pen/Strep (Gibco). Proliferation experiment in MiaPaCa2 cells was performed by plating at 10,000 cells/well and adding methyl acetate (2 or 10 mM), tert-butyl acetate (2 or 10 mM), or acetate (2 or 10 mM) the following day. Cells were fixed with 10% formalin after overnight culture to attach ('d0') and after five days of proliferation ('d5'). Relative cell proliferation was quantified by staining cells with crystal violet and quantifying relative dye quantity by absorbance at 595 nm of d5 and d0 timepoints after background correction. At least once prior to all experiments, cell lines were verified to be mycoplasma-free by PCR. All cell lines were authenticated by STR DNA fingerprinting within the last two years, and a central cell bank was maintained containing authenticated cell lines.

**Chemicals.** $^{13}C_5$-labeled glutamine (CLM-1822-H), $^2H_6$-labeled α-ketoglutaric acid (DLM-9476), $^{13}C_4$-labeled succinic acid (CLM-1571), and $^{15}N$-labeled alanine (NLM-454) were acquired from Cambridge Isotope Laboratories. HPLC-grade reagents, including methanol, chloroform, water, and acetone were acquired from Sigma. Oligomycin (Cayman Chemicals), FCCP (Cayman Chemicals), rotenone (Cayman Chemicals), antimycin A (Sigma), Seahorse XFe96 XF base media (Agilent), D-glucose (Sigma), sodium chloride (Sigma), L-glutamine (Sigma), sodium pyruvate (Sigma), α-ketoglutaric acid disodium salt (Sigma), dimethyl 2-oxoglutarate (Sigma), DBE-4 (dimethyl-succinate; Sigma), DBE-5 (dimethyl-glutarate; Sigma), 1-octyl-α-ketoglutarate (Cayman Chemicals), methyl pyruvate (Sigma), tert-butyl acetate (Sigma), sodium acetate (Sigma), L-alanine methyl ester hydrochloride (Sigma), L-alanine ethyl ester hydrochloride (Sigma), L-alanine tert-butyl ester hydrochloride (Alfa Aesar), sodium fumarate dibasic (Sigma), dimethyl fumarate (Sigma), diethyl fumarate (Sigma), norvaline (Sigma), deferoxamine mesylate (Cayman Chemicals), doxycycline hyclate (Sigma), hexadimethrine bromide (Sigma), hygromycin B (Sigma), methoxyamine hydrochloride (Sigma), and MTBSTFA + 1% TBDMSCI (Sigma).

**Metabolite extraction and GC-MS analysis.** For stable-isotope tracing and uptake experiments, cells were plated at 250,000 cells/well into six-well dishes and allowed to attach overnight. The following day, cells were washed with saline to remove media contaminants, and tracing media or DMEM containing methyl acetate (1 mM), KG (1 mM), or DMKG (1 mM) and 10% dialyzed serum was added. Tracing media was prepared from DMEM lacking sodium bicarbonate, phenol red, glucose, and L-glutamine (Sigma D5030) with 4 mM $^{13}C_5$-glutamine and 25 mM D-glucose and supplemented with 10% dialyzed serum and 3.7 g/L of sodium bicarbonate. Cells were cultured with tracing media or supplemented DMEM for 24 h and extracted by briefly washing cells with ice cold saline (0.9% NaCl prepared in HPLC-grade water) and adding 500 μl of HPLC-grade methanol and 200 μl of HPLC-grade water containing norvaline (1 μg per sample). Cells were scraped and transferred to a tube containing 500 μl of HPLC-grade chloroform, vortexed for 5–10 m, and centrifuged to separate polar and non-polar metabolite extracts. Polar metabolite extracts were dried under vacuum using a SpeedVac (Thermo Savant) and derivatized and analyzed by GC-MS as described below. MPE was quantified by taking a weighted average of the mass isotopologue distribution to quantify the total carbon contribution $^{13}C_5$-labeled glutamine and/or unlabeled KG to intracellular metabolites.

Cell-free hydrolysis experiments were performed in 500 μl of DMEM containing 1% Pen/Strep on a dry bath at 37 °C. At initial and specified time points, 5 μl was removed from the reaction and quenched in 250 μl of HPLC-grade methanol containing either 5 nmol of $^2H_6$-labeled KG and 5 nmol of $^{13}C_4$-labeled succinate or 5 nmol of $^{15}N$-labeled alanine depending on the ester analyzed. Fumarate ester hydrolysates were extracted similarly but in 250 μl of HPLC-grade methanol containing norvaline (1 μg per sample). Derivatization and GC-MS analysis were performed similar to described below; however, the GC temperature

was adjusted to the following parameters: 100 °C after injection, ramped to 320 °C at 17.5 °C/min, held at 320 °C for 3 m.

An anoxic environment was generated using a modified oxygen-controlled chamber (Coy Labs, O2 Control InVitro Glove Box) as previously described[55]. First, oxygen levels were decreased and maintained at 0.1% using pure nitrogen; $CO_2$ was maintained at 5% throughout. Oxygen and $CO_2$ sensors were calibrated every 3–4 months using a 2-step calibration process and prepared gas tanks at 5% $CO_2$ and 0.1% oxygen. Anoxic conditions were generated by flooding the chamber with a hydrogen gas mix (5% $CO_2$, 5% $H_2$, 90% $N_2$), and a palladium catalyst was added to eliminate excess oxygen and maintain levels at 0–5 ppm (<0.001% $O_2$). Oxygen levels were monitored using the anaerobic monitor (Coy Labs, CAM-12), and a humidified 37 °C environment was maintained by ambient heating and air circulation. For anoxic stable-isotope dilution experiments, 8988T cells were plated in six-well dishes at 250,000 cells/well and pre-incubated in anoxic environments for 72 h prior to assay. Tracing media containing $^{13}C_5$-glutamine and unlabeled KG (1 mM) or methyl acetate were prepared as described above and pre-incubated in the anoxic environment to degas oxygen. Cells were briefly washed with anoxic PBS and incubated with tracing media for 24 h. Metabolites were extracted similarly to normoxic conditions; however, all steps up to centrifugation of metabolite extracts were performed in anoxic conditions.

For deferoxamine-treated stable-isotope dilution experiments, 8988T cells were plated in six-well dishes at 250,000 cells/well and pre-treated with either DFO (500 µM) or vehicle (PBS). The following day, cells were washed with PBS and tracing media containing 4 mM $^{13}C_5$-glutamine treated with either DFO (500 µM) or vehicle (PBS) and/or KG (1 mM). Cells were incubated for 6 h and metabolites were extracted using methanol-water-chloroform containing norvaline (1 µg/sample) as described above.

Dried polar metabolite extracts were derivatized with 20 µl of methoxyamine hydrochloride (20 mg/ml in pyridine, prepared fresh) for 30–60 m at 37 °C and 20 µl MTBSTA + 1% TBDMSCl for 5–10 m at 37 °C and analyzed by GC-MS. GC-MS analysis was performed using an Agilent 7890B gas chromatograph (GC) with a DB-35MS column (30 m × 0.25 mm i.d. × 0.25 µm) installed and coupled to an Agilent 5977B mass spectrometer. The GC temperature was adjusted to the following parameters: 100 °C held after injection, ramped to 255 °C at 7.5 °C/min, ramped to 320 °C at 15 °C/min, held at 320 °C for 3 m, and post-run held at 320 °C for 2 m. The MS was operated in full-scan mode from 100 to 650 $m/z$. Mass isotopologue distributions of selected metabolite ion fragments were quantified and corrected for natural isotope abundance using algorithms adapted from ref. [93].

### Metabolite extraction and LC-MS analysis.
Extraction of metabolites from cell pellets— Metabolites were initially extracted from samples by quickly aspirating the cell culture media and adding 1 mL of extraction buffer, consisting of 80% methanol (Fisher Scientific) and 500 nM metabolomics amino acid mix standard (Cambridge Isotope Laboratories). To effectively scale all harvested samples to equivalent volumes of extraction buffer, samples were fully dried down by Speedvac (Thermo Fisher, Waltham, MA) and reconstituted volumetrically by mixing the entire dried cell pellet sample with 1 mL of 80% methanol without QC standards in 2.0 mL screw cap vials containing ~100 µL of disruption beads (Research Products International, Mount Prospect, IL). Samples were scaled to a ratio of 1e6 cells to 1 mL of extraction solvent with all steps being carried out in a cold room. Each was homogenized for 10 cycles on a bead blaster homogenizer (Benchmark Scientific, Edison, NJ). Cycling consisted of a 30 sec homogenization time at 6 m/s followed by a 30 sec pause. Samples were subsequently spun at $21,000 \times g$ for 3 min at 4 °C. A set volume of each (450 µL) was transferred to a 1.5 mL tube and dried down by Speedvac concentration. Samples were reconstituted in 50 µL of Optima LC/MS grade water (Fisher Scientific, Waltham, MA). Samples were sonicated for 2 min, then centrifuged at $21,000 \times g$ for 3 min at 4 °C. Twenty microliters were transferred to LC vials containing glass inserts for analysis. The remaining sample was placed at −80 °C for long-term storage.

Samples were subjected to an LC-MS analysis to detect and quantify known peaks. A metabolite extraction was carried out on each sample by quickly aspirating experimental media and adding 1 mL of 80% methanol containing internal QC standards. A Millipore™ ZIC-pHILIC (2.1 × 150 mm, 5 µm) LC column was coupled to a Dionex Ultimate 3000™ system. The column oven temperature and flow rate were set to 25 °C and 100 µL/min, respectively, for the following gradient elution: 80–20%B (0–30 min), 20–80%B (30–31 min), 80–80%B (31–42 min). Mobile phase compositions were: (A) 10 mM ammonium carbonate in water, pH 9.0 and (B) neat acetonitrile; and an injection volume of 2 µL was used for all analyses.

The LC system was coupled to a Thermo Q Exactive HF™ mass spectrometer operating in heated electrospray ionization mode (HESI) for LC-MS analysis. A 30-m polarity switching data-dependent Top 5 method was used for both positive and negative modes. The following parameters were also set: spray voltage of 3.5 kV, capillary temperature at 320 °C, sheath gas flow rate of 35, aux gas rate of 10, and max spray current of 100 µA. Full MS scan parameters for both positive and negative modes were set as followed: scan range of 67–1000 $m/z$, resolution of 120,000, AGC target of 3e6, and maximum IT of 100 ms. Tandem MS spectra for both positive and negative modes used a resolution of 15,000, fixed first mass of 50 $m/z$, isolation window of 0.4 $m/z$, isolation offset of 0.1 $m/z$, AGC target of 1e5, minimum AGC target of 1e4, intensity threshold of 2e5, maximum IT of 50 ms,

and three-way multiplexed normalized collision energies (nCE) of 10, 30, 80. All data were acquired in profile mode.

Thermo™ RAW files were read using ThermoFisher CommonCore RawFileReader. An in-house python script (Skeleton) was used for detection and quantification of sample peaks and internal standards based on a retention time and accurate mass library adapted from the Whitehead Institute[94] and verified with authentic standards and/or high-resolution MS/MS spectra manually curated using the NIST14MS/MS[95] and METLIN (2017)[96] tandem mass spectral libraries. For feature-based analysis, an in-house python script (Ungrid) was used to detect MS1 peaks across all samples using the following parameters: a $m/z$ discrimination threshold of 20 ppm, a minimum peak intensity of 1e5, a minimum signal-to-noise ratio of 10, and a retention time threshold of 2 min. Metabolite and feature peaks extracted in this manner were defined by either the detected feature $m/z$ or the theoretical $m/z$ of the expected ion type for the standard in the library (e.g., $[M+H]^+$). The following parameters were applied: a ±5 part-per-million (ppm) tolerance, an initial retention time search window of ±0.5 min across all samples, and a ±7.5 s peak apex retention time tolerance within individual samples. An in-house statistical pipeline, Metabolize (version 1.0), was used to process the resulting data matrix of metabolite intensities for all samples and blank controls. A final peak detection was calculated based on a signal-to-noise ratio (S/N) or 3× blank controls with a floor of 1e5 (arb. units). The threshold value was input for any sample where the calculated peak intensity was lower than the blank threshold for any statistical comparisons. The resulting blank corrected data matrix was used for all group-wise comparisons. $T$-tests were performed using the Python SciPy library (version 1.1.0)[97] to test for differences and generate statistics. Any metabolite with $p$-value < 0.05 was considered significantly regulated (up or down). Volcano plots were generated utilizing Prism (GraphPad). The R package DESeq2 (1.24.0)[98] was used to adjust for covariate effects (as applicable) and to calculate the adjusted $p$-value in the covariate model. Zero values were input for non-detected values instead of the blank threshold to avoid false positive.

### Generation of inducible 2HG reporter Cells.
pSLIK-IDH1-R132H-FLAG (Addgene #66803) and pSLIK-IDH2-R172K-FLAG (Addgene #66807) were generously provided by Thekla Cordes, Esther Lim, and Christian Metallo. Lentivirus was produced by transfecting 293T cells with pSLIK, pMD2.G (Addgene #12259), and psPAX2 (Addgene #12260) using standard Lipofectamine 3000 (Thermo) protocol. Viral supernatant was collected after 48 and 72 h, filtered with 0.45 µm pore size filters, and used fresh or frozen at −80 °C for long-term storage. HuPT4 and 8988S pSLIK cells were generated by plating in six-well plates with 1 mL of media containing 0.5 µg/ml of hexadimethrine bromide and infecting with 2 mL of virus supernatant. Cells were selected using hygromycin (500 µg/ml) for ~7–10 days. Cells were considered fully selected by comparing to selection of non-infected surrogate plate. Induction of IDH1-R132H or IDH2-R172K was performed in DMEM containing dialyzed FBS and 0.5 µg/mL doxycycline for 24 h. 2-HG levels and mass isotopologue distributions were measured by GC-MS as described above[57].

### Oxygen consumption rate (OCR) and extracellular acidification rate (ECAR) determination.
For determination of glycolytic and respiratory effects in response to stimulation with metabolite or esterified analogs, 8988T cells were plated at 15,000 cells/well in 96-well assay plates (Agilent) in DMEM containing 10% dialyzed FBS and 1% Pen/Strep and allowed to attach overnight. The Seahorse XFe96 assay cartridge (Agilent) was hydrated in 200 µl of ddH₂O in a non-CO₂ 37 °C incubator the day prior to assay, switched to 200 µl of prewarmed calibrant solution (Agilent) the next day, and incubated in a non-CO₂ 37 °C incubator at least 1 h prior to assay. XF assay media was prepared by adding 20 mM of D-glucose and 2 mM of L-glutamine to XF base media, which contains HEPES (5 mM) (Agilent). Prior to assay, 8988T cells were washed with XF assay media twice and cells were incubated in a non-CO₂ incubator at 37 °C for ~45–60 m in 150 µl of assay media. Injection port A was loaded with 25 µl of assay media containing 7 mM (7× concentration) of metabolite or esterified analogs or methyl acetate as a vehicle control. Basal ECAR and OCR were established for ~18 m followed by injection of port A. Dynamics of glycolytic and respiratory metabolism following the ~60-m stimulation was assayed by sequential injection of oligomycin (1 µM), FCCP (0.5 µM), and rotenone/antimycin (1 µM/1 µM). Cell-free acidification was assayed alongside cell-based measurements following similar protocol but excluding cells. Both cell-free and bioenergetic (OCR/ECAR) measurements were made using a modified DMEM containing a HEPES buffer (5 mM). At least three independent experiments were performed for all conditions, and data from 6 to 12 technical replicate wells for each independent experiment were averaged and plotted as mean ± s.d. for each experimental group. Data were normalized to the average of each basal rate and plotted as a percentage.

### Reporting summary.
Further information on research design is available in the Nature Research Reporting Summary linked to this article.

## Data availability
The LC-MS data generated in this study have been deposited in the National Metabolomics Data Repository (https://www.metabolomicsworkbench.org/) under study ID ST001860. The processed GC-MS data generated in this study are provided in the

Source Data file. METLIN (2017) is available from https://www.sisweb.com/software/ms/wiley-metlin.htm (Accessed on July 10, 2021). Other data associated with this study are available from the corresponding author(s) upon reasonable request. Source data are provided with this paper.

## Code availability
Metabolyze is available at https://github.com/DrewRJones/Metabolyze_Public.git (Accessed on July 10, 2021) under MIT license and requires Python 3.6.5 and R 3.5.1. GC-MS analysis script is available at https://github.com/Sethjparker/IntegrateNetCDF_WithCorrect (Accessed on July 10, 2021) under MIT license and requires Matlab R2016 or higher.

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

## Acknowledgements

This work was supported by National Cancer Institute Grants R01CA157490, R01CA188048, P01CA117969, R35CA232124; ACS Research Scholar Grant RSG-13-298-01-TBG; NIH grant R01GM095567; P30CA016087; and the Lustgarten Foundation, and SU2C to A.C.K.; S.J.P. was supported by American Cancer Society grant 132942-PF-18-215-01-TBG. The authors would like to thank Shoukat Dedhar for discussions related to cellular pH regulation. The authors would also like to thank Diana Liaw for help with figure preparation.

## Author contributions

S.J.P. and A.C.K. conceptualized the project, designed all experiments, and wrote the manuscript; K.E.R.H., J.E.R., J.A.K.R., E.Y.L., and A.S.W.S. conducted biochemical experiments; D.M.H. provided essential chemistry expertise and aided in the design of biochemical experiments; L.J.A. and D.R.J. assisted with LC-MS acquisition and analysis; M.R.P. provided essential support for the acquisition of data. All authors participated in critical review of the manuscript.

## Competing interests

A.C.K. has financial interests in Vescor Therapeutics, LLC. A.C.K. is an inventor on patents pertaining to KRAS-regulated metabolic pathways, redox control pathways in pancreatic cancer, targeting GOT1 as a therapeutic approach, and the autophagy control of iron metabolism. A.C.K. is on the SAB of Rafael/Cornerstone Pharmaceuticals. A.C.K. is a consultant for Deciphera and Abbvie. The other authors declare no competing interests.
