## [Peer Review File · Nature Communications]

Reviewers' Comments:

Reviewer #1:

Remarks to the Author:

This is an interesting paper demonstrating esterified KG analogs rapidly hydrolyze in aqueous media, yielding KG. Much of the work is well done. Overall the paper will be of interest to the metabolism community.

However, I completely disagree with their conclusion it has minimal effects on TCA cycle. The key experiment that was done by multiple labs was to prevent α KG into the TCA cycle by giving AOA (aminotransferase inhibitor) or glutaminase and examining OCR. In every case, it was shown that at DMK can increase OCR in the presence of AOA or glutaminase. Please conduct this experiment with all the esters. Examine OCR (Basal and coupled) and ECAR with AOA +/- different esters. I would not expect DMK to have any effect (exactly what the paper contends) without AOA or glutaminase inhibition. DMK rescues glutaminolysis inhibition!

Please discuss why your results differ from Eyal Gottlieb's previous work in MCB 2007 (Cell-Permeating α -Ketoglutarate Derivatives Alleviate Pseudohypoxia in Succinate Dehydrogenase-Deficient Cells).

Excerpt from the paper: "When cells were treated with octyl- α -ketoglutarate or TFMB- α -ketoglutarate for 2 h prior to extraction, intracellular α -ketoglutarate levels rose by approximately fourfold (Fig. 2c). In contrast, cells treated with either underivatized or benzyl- α -ketoglutarate (the least hydrophobic derivative) (Table 1) showed no increase in the level of intracellular α -ketoglutarate above basal levels (Fig. 2c). These results indicate that underivatized α -ketoglutarate does not efficiently enter cells, and this emphasizes that derivatizing the acid, and introducing sufficient hydrophobicity, is crucial for effectiveness. In subsequent experiments, octyl- and TFMB- α -ketoglutarate esters were used to elevate intracellular α -ketoglutarate, while underivatized and benzyl- α -ketoglutarate were used as negative controls".

Reviewer #2:

Remarks to the Author:

The manuscript by Parker et al provides a detailed examination of the chemical and metabolic properties of 2-ketoglutarate and related metabolite esters. These reagents are commonly applied as delivery vehicles to determine the causative properties of the parent metabolite in cell culture studies, but many facets of their activity are surprisingly uncharacterized. The authors use careful LC-MS driven metabolomics to characterize the properties of KG and its esters, leading to the identification of several unintended properties of the latter including rapid hydrolysis, extracellular acidification, and metabolic perturbations. While this data will be undoubtedly useful to the metabolism community, what I found even more exciting was that they used their in-depth methods to generate evidence that 1) KG salts can be taken up by some cell lines, and 2) extracellular KG are primarily utilized in the cytosol, suggesting that distinct subcellular KG pools which respond uniquely to external stimuli. In my view this latter finding is the most exciting of the paper, and should be highlighted in the abstract (caveat: it is possible this may not be as novel as it seems to me, and leave it to the authors to couch it properly in the context of the literature which they no doubt no better). This manuscript provides an impactful resource, and I have only a few recommendations so that it might be further improved:

1. The authors should be sure to emphasize in their conclusion the considerable heterogeneity observed in 2-KG uptake between different cell lines. This is important, as it will stress the need for researchers to quantify delivery (using assays such as the competitive glutamine labeling experiment described here) prior to interpreting results from extracellular KG/KG ester administration.

2. Another useful qualification of the authors findings that should be emphasized in the summary is that a single concentration is examined in this study, and the comparisons of KG and KG esters may differ depending on the concentration at which they are administered. It is in theory possible that at lower concentrations, KG esters may 'outperform' non-esterified KG and deliver higher

effective molarities of KG to the cytosol or mitochondria. Alternatively, it is possible that at low concentrations KG esters may prove too inefficient to manifest any studies KG-dependent effects at all. Consistent with the latter view, two previous studies have provided evidence that at low concentrations KG diesters act as inhibitors of KG-dependent dioxygenases, presumably due to slow hydrolysis of one of the ester moieties (Hou et al., PLOS One 2014, doi: 10.1371/journal.pone.0113865, Zengeya et al Org Lett 2015, 10.1021/acs.orglett.5b00737). I note that I am biased a little, since I am Meier, one of the authors of the latter paper. However, it would be resonant with your hypothesis that esterified KG (and downstream metabolites) and could provide some useful context for non-experts.

Minor points

3. In the experiment in Figure 1f-g the authors demonstrate the ability of extracellular KG to form a large proportion of the intracellular KG pool (as compared to glutamine), raising the question while reading as to whether the overall pool is larger. The authors demonstrate this much later in the paper, but it would be helpful if they alluded to this at the point in the text where they introduce this data (i.e. "Extracellular KG or DMKG comprised approximately 40-80% of the intracellular pool, depending on cell line, suggesting that uptake can act as a major source of intracellular KG (Fig. 1g) and increase overall levels of cellular KG (vide infra)."

4. On page 13, it would be helpful in introducing this experiment if the authors specified the subcellular localization of IDH1/2 to better describe how it differentiates mitochondrial/extramitochondrial metabolism.

5. "Anoxic culture conditions were achieved by depletion of oxygen using a palladium catalyst to maintain oxygen levels at 3-5 ppm." I was not aware of this chemical method for oxygen depletion in cell culture and it sounds really cool, could the authors provide a reference?

6. The regioselective non-enzymatic hydrolysis of the 1-position of 2-KG esters due to its increased leaving group potential was also described in the above mentioned Zengeya et al. reference (Org Lett 2015, 10.1021/acs.orglett.5b00737) and may have some discussion points useful for the revision.

7. One part of the paper manuscript I found a little counterintuitive was the differential effects of ketoacid esters (acid pKa~2.5) versus traditional esters (acid pKa ~4.5) on media pH. At physiological pHs both of the released acids would be >90% deprotonated and therefore reasonably expected to contribute similarly to acidification. One explanation for the difference would be if, at 1 mM, the released acids exceeded the buffering capacity of the medium. Could the authors better explain this phenomena, incorporating a reference to the buffering capacity of the media?

9. A recent study by Swain et al. (<https://doi.org/10.1038/s42255-020-0210-0>) described the distinct properties of itaconate and itaconate esters that have been commonly used in immunometabolism studies. There is some resonance with the authors findings, in that Swain et al. found non-esterified itaconate was efficiently delivered and that itaconate esters have unintended consequences. It would be great if a link could be drawn between these two studies in the discussion, and I am sure the other authors would appreciate it too.

Again, overall this is a beautiful paper and I look forward to reading the final version in Nature Communications. Congratulations!

Jordan Meier

Reviewer #3:

Remarks to the Author:

Review of paper by Parker et al.

The authors explore the effects of α KG esters versus "pure" analogs on cellular metabolism and acidification. They demonstrate that rapid hydrolysis occurs in cell media independent of esterases and induce significant α KG-independent effects on cellular metabolism. The present study is timely and highlights important differences between cell-permeable esterified analogs and their parent compounds, but further experimentation is required to support some of the findings.

1. Spontaneous hydrolysis should be examined over a shorter time course. The earliest time point examined is 24 h. How quickly does hydrolysis occur?
2. Similarly, intracellular α KG levels should be assessed at earlier time points. Do α KG and ester analogs increase intracellular α KG at the same rate? This should be assessed over the same time course as the functional seahorse experiments. It is important to determine if the metabolic effects are observed because the ester analog generates intracellular α KG on a faster time course to pure α KG.
3. The effects of all compounds on cellular toxicity at these long time points should be assessed.
4. Given the vast array of literature using esterified analogs in immune cells it would be interesting to repeat these experiments in an immune cell type.
5. The authors state that the acidification driven by DM-KG and 1-octyl-KG is responsible for the decrease in ECAR but only DM-KG decreases ECAR, not 1-octyl-KG. How do the authors reconcile this difference?
6. The acidification assay should be performed in a cellular system, which would likely be buffered. Do these compounds still decrease pH in cells? Or across the cell membrane? There are established microscopy methods to measure pH in live cells.
7. How does DMKG affect TCA metabolism? This is unclear. These comparators need to be performed when the absolute levels of α KG derived from exogenous α KG and ester analogs are the same. The time point here was 3 h. What do the absolute levels of α KG look like at this time point?

We are delighted that *Nature Communications* will consider a revised version of our manuscript by Parker et al. titled “Spontaneous hydrolysis and spurious metabolic properties of α -ketoglutarate esters”. We were pleased with the constructive comments by the reviewers, which have helped to guide the revision and strengthen the manuscript.

Please find below a point-by-point response to the reviewer’s concerns (in blue). Specific changes to the manuscript that incorporate reviewer suggestions and new data are underlined below.

Sincerely,

Alec Kimmelman and Seth Parker

REVIEWER COMMENTS

Reviewer #1 (Remarks to the Author):

This is an interesting paper demonstrating esterified KG analogs rapidly hydrolyze in aqueous media, yielding KG. Much of the work is well done. Overall the paper will be of interest to the metabolism community.

However, I completely disagree with their conclusion it has minimal effects on TCA cycle. The key experiment that was done by multiple labs was to prevent aKG into the TCA cycle by giving AOA (aminotransferase inhibitor) or glutaminase and examining OCR. In every case, it was shown that at DMK can increase OCR in the presence of AOA or glutaminase. Please conduct this experiment with all the esters. Examine OCR (Basal and coupled) and ECAR with AOA +/- different esters. I would not expect DMK to have any effect (exactly what the paper contends) without AOA or glutaminase inhibition. DMK rescues glutaminolysis inhibition!

We agree with the reviewer that this point could be discussed further in light of reports by multiple labs, and we agree with the reviewer that our data does not support claims that KG cannot contribute to the TCA cycle. An important caveat is that our experiments were done in normal, unstressed conditions. In conditions where mitochondrial KG production is suppressed, such as during transaminase inhibition (AOA), glutaminase inhibition (CB-839, BPTES), or glutamine withdrawal, cytosolic/extracellular KG may contribute more significantly to mitochondrial metabolism. We have included additional discussion surrounding this point and have performed the requested Seahorse experiments and additional stable-isotope dilution studies using DMKG (see below):

Lines 10-12: “In many cell lines, imported KG was metabolized to succinate in the cytosol, and we observed minimal KG utilization for mitochondrial metabolism in normal culture conditions.”

Lines 290-303: “To determine if imported KG or DMKG could act as an anaplerotic source for mitochondrial TCA cycle metabolism, we quantified ^{13}C label-dilution into downstream TCA cycle intermediates (e.g., fumarate, malate, glutamate, and citrate) in HuPT4, 8988S, 8988T, and AsPC1 cell lines as well as RAW246.7 macrophages, given the broad usage of KG analogs in immune cell types (Fig. 5f). While we observed significant incorporation of KG into succinate pools across all cell lines except for RAW246.7 macrophages, we quantified

minimal contribution into TCA cycle metabolites (0.5-2%) in all KG supplemented cells (Fig. 5g, h; Extended Data Fig. 5b). To determine if this was a cancer cell line phenomenon, we repeated the stable-isotope dilution in a non-transformed mouse embryonic fibroblast (MEF) cell line, and we observed similar import of KG and decarboxylation to succinate but minimal contribution to TCA cycle metabolites (~2-3%) (Extended Data Fig. 5c)⁵⁶. These data suggest that KG import is context-dependent and does not significantly contribute to mitochondrial metabolism in normal culture conditions. Thus, the succinate derived from imported KG is likely independent of KGDH and is extra-mitochondrial.

Lines 351-353: “These data confirm that KG transport into mitochondria from the cytosol is limited, and that the known mitochondrial KG antiporter (SLC25A11) predominantly operates as a KG exporter in normal culture conditions (Fig. 6a).”

Lines 400-402: “Our data suggests that extracellular KG acts primarily as a substrate for cytosolic KG-dependent reactions and contributes minimally to mitochondrial TCA cycle metabolism in the cell lines examined and under standard culture conditions.”

As the reviewer suggests, we have also conducted Seahorse experiments investigating effects on OCR in response to AOA treatment +/- KG or DMKG. Following injection of AOA, OCR acutely decreased by ~40% over 60 minutes (Reviewer Fig. 1A). However, injection of either 1 mM KG or DMKG failed to rescue AOA-induced OCR defects in four independent experiments (Reviewer Fig. 1A, B). Although this has been shown in multiple studies, in our hands 8988T pancreatic cancer cells do not exhibit a similar rescue suggesting a context-dependent effect, which we have discussed further in the revised manuscript. Notably, PDAC cells rely on mitochondrial GOT2 for aspartate production and proliferation (PMID: 23535601, 33027658, 33230296), and KG or DMKG supplementation would not be sufficient to rescue the on-target GOT2 inhibitory effects of AOA.

Reviewer Figure 1. AOA treatment acutely decreases OCR in 8988T cells. Injection with 1 mM of KG or DMKG fails to significantly rescue AOA-induced OCR defects relative to vehicle (methyl acetate).

In light of these findings, we have conducted stable-isotope tracer dilution studies using DMKG, which is commonly used to rescue inhibition of glutamine metabolism as the reviewer points out. Unlike KG as shown in our initial manuscript, we found that DMKG contributes significantly to TCA cycle intermediates in many cell lines (Reviewer Fig. 2A). As further support, our earlier manuscript showed that methylated TCA intermediates (methyl-succinate, methyl-fumarate) accumulated in DMKG-treated cells, suggesting that mono-methylated KG might retain mitochondrial membrane permeable properties and contribute to the TCA cycle directly. We believe these new data are relevant to our study and provide mechanistic evidence as to

why/how DMKG may rescue mitochondrial metabolism defects in other published contexts. We have included these new data in the revised manuscript (Revised Fig. 5g, h), and have included the following discussion points (see below).

Reviewer Figure 2. (A) DMKG contributes significantly to mitochondrial TCA cycle intermediates, whereas KG does not. (B & C) DMKG treatment results in accumulation of mono-methylated succinate and fumarate in 8988T cells treated for 3-hours.

Lines 419-439: “While KG and esterified analogs both contributed to intracellular succinate pools, indicative of cytosolic/nuclear dioxygenase activity, DMKG also uniquely contributed significantly to mitochondrial TCA cycle metabolism. These results suggest that mitochondrial membranes are permeable to methylated KG, likely as the mono-methylated form given the speed at which hydrolysis of the proximal α -ketoester occurred. This is further supported by the accumulation of methylated TCA intermediates in DMKG-treated cells, including methyl-succinate and methyl-fumarate, which are presumably generated by metabolism of mono-methyl-KG by mitochondrial TCA cycle enzymes. Dietary sources of esterified TCA intermediates exist, including diethyl-malate from fermented products such as beer and dimethyl-succinate from peanuts and roasted filberts, and may have “exposomic” effects on (patho)physiology⁶⁹. However, DMKG did not contribute to mitochondrial TCA cycle metabolites in RAW246.7 macrophages, which may suggest that specific transporter(s) are required for mitochondrial import of 5MKG/DMKG. This is similar to a recent report by Fets et al., which found that monocarboxylate transporter 2 (MCT2) was required for the import of methyloxalylglycine (MOG), a mono-methyl ester derived from the hydrolysis of dimethyloxalylglycine (DMOG) that is commonly used to inhibit prolyl hydroxylase (PHD) activity and stabilize HIF1 α ³³. Together, these data provide mechanistic evidence for how DMKG may rescue mitochondrial metabolism defects that arise from glutamine withdrawal and glutaminase or transaminase inhibition in other published contexts^{13-15,21,22,24,26,27,29}. However, it is important to understand whether this contribution is unique to esterified KG, or if endogenously produced and/or supplemented KG can import into the mitochondrial matrix when substrates like glutamine are limited, which requires further study.”

Please discuss why your results differ from Eyal Gottlieb's previous work in MCB 2007 (Cell-Permeating α -Ketoglutarate Derivatives Alleviate Pseudohypoxia in Succinate Dehydrogenase-Deficient Cells).

Excerpt from the paper: "When cells were treated with octyl- α -ketoglutarate or TFMB- α -ketoglutarate for 2 h prior to extraction, intracellular α -ketoglutarate levels rose by approximately fourfold (Fig. 2c). In contrast, cells treated with either underivatized or benzyl- α -ketoglutarate (the least hydrophobic derivative) (Table 1) showed no increase in the level of intracellular α -ketoglutarate above basal levels (Fig. 2c). These results indicate that underivatized α -ketoglutarate does not efficiently enter cells, and this emphasizes that derivatizing the acid, and introducing sufficient hydrophobicity, is crucial for effectiveness. In subsequent experiments, octyl- and TFMB- α -ketoglutarate esters were used to elevate intracellular α -ketoglutarate, while underivatized and benzyl- α -ketoglutarate were used as negative controls".

We thank the reviewer for pointing out this important study by MacKenzie et al. (MCB 2007, ref #11), which we cited in our original manuscript. The cell lines used in these studies differ, and we demonstrate in Figure 1D that KG accumulation significantly depends on the cellular context. For example, DanG pancreatic cancer cells and RAW246.7 macrophages did not accumulate KG to similar extents as other cell lines, and HEK293, HeLa, and HCT116 cells were not assayed in this study nor was DMKG evaluated in MacKenzie et al. Furthermore, our assays use a direct, mass spectrometry-based approach (in some contexts isotopically-labeled standards) to quantify absolute KG levels across multiple orthogonal biological and independent experiments and instruments. This approach differs from the enzyme-coupled glutamate dehydrogenase assays used by MacKenzie et al., and mass spectrometry likely offers improved sensitivity. We have included additional discussion to include studies by MacKenzie et al., Tennant et al., and Zengeya et al. that quantify KG and KG analog uptake by coupled enzyme assay in HEK293, HeLa, and HCT116 cells (see below).

Lines 388-398: "Not all of the cell types evaluated in our study were able to take up KG in its endogenous form. For example, supplemented KG did not significantly contribute to intracellular KG pools in RAW264.7 macrophages (Fig. 5g), suggesting a potential lack of KG transporter expression. Previous studies by MacKenzie et al., Tennant et al., and Zengeya et al. show that HEK-293, HCT116, and HeLa cells do not significantly take up KG and require octyl- and/or TFMB- modified analogs to increase intracellular KG levels and cause biological effects^{11,28,30}. We also observed considerable heterogeneity in KG uptake across the cell lines tested (Fig. 1d), emphasizing the need to carefully quantify uptake capacity and metabolic utilization using stable-isotope labeling in each context. How KG is transported by cancer, immune, and other cell types is not well understood and warrants future study."

Lines 449-455: "It is possible that esterified analogs may outperform endogenous KG uptake at lower concentrations and/or have distinct biological effects depending on the context and mechanism of KG uptake. Previous studies using TFMB-, octyl-, and/or dimethyl- KG analogs report opposing effects on HIF1 α stabilization—via modulation of the activity of prolyl hydroxylases (PHDs)—in different cellular contexts, conditions, and KG analog concentrations^{11,28,30,31}. Further, TFMB alcohol alone, released during hydrolysis of TFMB-KG analogs, has also been shown to stabilize HIF1 α ³⁰."

Reviewer #2 (Remarks to the Author):

The manuscript by Parker et al provides a detailed examination of the chemical and metabolic properties of 2-ketoglutarate and related metabolite esters. These reagents are commonly applied as delivery vehicles to determine the causative properties of the parent metabolite in cell culture studies, but many facets of their activity are surprisingly uncharacterized. The authors use careful LC-MS driven metabolomics to characterize the properties of KG and its esters, leading to the identification of several unintended properties of the latter including rapid hydrolysis, extracellular acidification, and metabolic perturbations. While this data will be undoubtedly useful to the metabolism community, what I found even more exciting was that they used their in-depth methods to generate evidence that 1) KG salts can be taken up by some cell lines, and 2) extracellular KG are primarily utilized in the cytosol, suggesting that distinct subcellular KG pools which respond uniquely to external stimuli. In my view this latter finding is the most exciting of the paper, and should be highlighted in the abstract (caveat: it is possible this may not be as novel as it seems to me, and leave it to the authors to couch it properly in the context of the literature which they no doubt no better). This manuscript provides an impactful resource, and I have only a few recommendations so that it might be further improved:

We would like to thank the reviewer for their enthusiasm for our study and for the excellent suggestions. We agree that the finding that extracellular KG is primarily used in the cytosol is very exciting and have revised the abstract to highlight this observation (see below):

Lines 2-14: “ α -ketoglutarate (KG), also referred to as 2-oxoglutarate, is a key intermediate of cellular metabolism with pleiotropic functions. Cell-permeable esterified analogs are widely used to study the role of KG in governing bioenergetic and amino acid metabolism and DNA, RNA, and protein hydroxylation reactions, as cellular membranes are thought to be impermeable to KG. Here we show that esterified KG analogs rapidly hydrolyze in aqueous media, yielding KG that, in contrast to prevailing assumptions, can be imported by many cell lines. Esterified KG analogs exhibited spurious KG-independent effects on cellular metabolism, including extracellular acidification, arising from rapid hydrolysis and de-protonation of α -ketoesters, and significant analog-specific inhibitory effects on glycolysis or mitochondrial respiration. In many cell lines, imported KG was metabolized to succinate in the cytosol, and we observed minimal KG utilization for mitochondrial metabolism in normal culture conditions. These findings demonstrate that nuclear and cytosolic KG-dependent reactions may derive KG from functionally distinct subcellular pools and sources.”

1. The authors should be sure to emphasize in their conclusion the considerable heterogeneity observed in 2-KG uptake between different cell lines. This is important, as it will stress the need for researchers to quantify delivery (using assays such as the competitive glutamine labeling experiment described here) prior to interpreting results from extracellular KG/KG ester administration.

We agree with the reviewer have included additional emphasis of this finding (see below):

Lines 388-398: “Not all of the cell types evaluated in our study were able to take up KG in its endogenous form. For example, supplemented KG did not significantly contribute to intracellular KG pools in RAW264.7 macrophages (Fig. 5g), suggesting a potential lack of KG transporter expression. Previous studies by MacKenzie et al., Tennant et al., and Zengeya et al. show that HEK-293, HCT116, and HeLa cells do not significantly take up KG and require octyl- and/or TFMB- modified analogs to increase intracellular KG levels and cause biological effects^{11,28,30}. We also observed considerable heterogeneity in KG uptake across the cell lines tested (Fig. 1d), emphasizing the need to carefully quantify uptake capacity and metabolic utilization using stable-

isotope labeling in each context. How KG is transported by cancer, immune, and other cell types is not well understood and warrants future study.”

2. Another useful qualification of the authors findings that should be emphasized in the summary is that a single concentration is examined in this study, and the comparisons of KG and KG esters may differ depending on the concentration at which they are administered. It is in theory possible that at lower concentrations, KG esters may ‘outperform’ non-esterified KG and deliver higher effective molarities of KG to the cytosol or mitochondria. Alternatively, it is possible that at low concentrations KG esters may prove too inefficient to manifest any studies KG-dependent effects at all. Consistent with the latter view, two previous studies have provided evidence that at low concentrations KG diesters act as inhibitors of KG-dependent dioxygenases, presumably due to slow hydrolysis of one of the ester moieties (Hou et al., *PLoS One* 2014, doi: 10.1371/journal.pone.0113865, Zengeya et al *Org Lett* 2015, 10.1021/acs.orglett.5b00737). I note that I am biased a little, since I am Meier, one of the authors of the latter paper. However, it would be resonant with your hypothesis that esterified KG (and downstream metabolites) and could provide some useful context for non-experts.

We found the studies by Zengeya et al., Hou et al., MacKenzie et al., and Tennant et al. to be highly influential in our revision of this manuscript, specifically when discussing the context dependence of KG uptake and KG analog effects (see below). We apologize for missing some of these important papers in our initial manuscript.

Lines 440-455: “Further, it is not well understood how physiological sources of KG may influence metabolism and/or dioxygenase activity. Plasma and cerebrospinal fluid levels of KG in healthy children and adults are ~10 μ M and ~5 μ M, respectively, and elevated plasma KG levels (~20-60 μ M) have been reported in diabetic or pyruvate carboxylase deficient children^{70,71}. We use a super-physiological concentration of 1 mM in this study to represent the concentrations of DMKG and other analogs used in published studies, which typically range from 1-4 mM up to 32 mM. However, these high concentrations may enrich for negative characteristics associated with KG esters, including acidification and effects induced by monoesters and/or the released alcohol(s). Minimizing analog concentrations may be preferred to reduce these impacts and to better match physiological conditions. It is possible that esterified analogs may outperform endogenous KG uptake at lower concentrations and/or have distinct biological effects depending on the context and mechanism of KG uptake. Previous studies using TFMB-, octyl-, and/or dimethyl- KG analogs report opposing effects on HIF1 α stabilization—via modulation of the activity of prolyl hydroxylases (PHDs)—in different cellular contexts, conditions, and KG analog concentrations^{11,28,30,31}. Further, TFMB alcohol alone, released during hydrolysis of TFMB-KG analogs, has also been shown to stabilize HIF1 α ³⁰.”

Minor points

3. In the experiment in Figure 1f-g the authors demonstrate the ability of extracellular KG to form a large proportion of the intracellular KG pool (as compared to glutamine), raising the question while reading as to whether the overall pool is larger. The authors demonstrate this much later in the paper, but it would be helpful if they alluded to this at the point in the text where they introduce this data (i.e. “Extracellular KG or DMKG comprised approximately 40-80% of the intracellular pool, depending on cell line, suggesting that uptake can act as a major source of intracellular KG (Fig. 1g) and increase overall levels of cellular KG (vide infra).”

Thank you for the suggestion. We included the intracellular KG level data earlier in the manuscript (Fig. 1d), but these results were not clearly written in the original manuscript. We have revised this section to help clarify (see below).

Lines 85-102: “To explore this, we cultured a panel of pancreatic cancer cell lines with either methyl acetate (vehicle), KG, or DMKG for 24 hours. In all cell lines tested, we observed a significant accumulation of KG when cells were supplemented with either KG or DMKG, suggesting that KG can diffuse into cells (Fig. 1d). KG accumulated to similar extents in KG- or DMKG- treated conditions with considerable heterogeneity across cell lines, suggesting that specific transporter(s) may be required to facilitate the import of KG (Fig 1d, e). To understand if extracellular KG could assimilate into intracellular metabolite pools, we conducted a reverse ¹³C-labeling experiment whereby intracellular KG, synthesized predominantly from glutamine and subsequent deamination of glutamate, was labeled using ¹³C₅-glutamine prior to the addition of unlabeled (¹²C) KG or DMKG (Fig. 1e). As expected, glutamine represented the major carbon source for synthesized KG contributing ~75-92% of carbon across cell lines as quantified by the mole percent enrichment (MPE) (Fig 1f, g). After 24 hours, we observed a significant dilution of labeled KG species (M1-M5) by unlabeled (M0) in 8988S, AsPC1, and 8988T cell lines arising from import and assimilation into intracellular KG pools (Fig. 1f; Extended Data Fig. 1b). Extracellular KG or DMKG comprised approximately 40-80% of the intracellular pool, depending on cell line, suggesting that uptake can increase cellular levels (Fig. 1d) and provide a major source of intracellular KG (Fig. 1g).”

4. On page 13, it would be helpful in introducing this experiment if the authors specified the subcellular localization of IDH1/2 to better describe how it differentiates mitochondrial/extramitochondrial metabolism.

We agree (see below):

Lines 337-339: “To quantify the compartment-specific KG metabolism, we used a previously described reporter system capable of differentiating cytosolic and mitochondrial labeling of 2-hydroxyglutarate (2-HG), the product of IDH1^{R132H/+} (cytosolic) and IDH2^{R172K/+} (mitochondrial) mutant enzymes⁵⁷.”

5. “Anoxic culture conditions were achieved by depletion of oxygen using a palladium catalyst to maintain oxygen levels at 3-5 ppm.” I was not aware of this chemical method for oxygen depletion in cell culture and it sounds really cool, could the authors provide a reference?

We share the reviewer’s enthusiasm for this method for depleting oxygen in cell culture! These approaches were developed by co-author Hollinshead, K.E.R. and were optimized and published in a recent study (Hollinshead et al. 2020. Cell Reports. 10.1016/j.celrep.2020.108231; ref #55). We have added additional reference to this approach in the revised manuscript and have detailed the approach in the revised methods (see below):

Lines 287-288: “Anoxic culture conditions were achieved by depletion of oxygen using a palladium catalyst to maintain oxygen levels at 3-5 ppm (<0.001% O₂) in a sealed glove box, as previously⁵⁵.”

Lines 561-569 (methods): “An anoxic environment was generated using a modified oxygen-controlled chamber (Coy Labs, O₂ Control InVitro Glove Box) as previously described⁵⁵. First, oxygen levels were decreased and maintained at 0.1% using pure nitrogen; CO₂ was maintained at 5% throughout. Oxygen and CO₂ sensors were calibrated every 3-4 months using a 2-step calibration process and prepared gas tanks at 5% CO₂ and 0.1% oxygen. Anoxic conditions were generated by flooding the chamber with a hydrogen gas mix (5% CO₂, 5% H₂, 90% N₂), and a palladium catalyst was added to eliminate excess oxygen and maintain levels at 0-5ppm

(<0.001% O₂). Oxygen levels were monitored using the anaerobic monitor (Coy Labs, CAM-12), and a humidified 37°C environment was maintained by ambient heating and air circulation.”

6. The regioselective non-enzymatic hydrolysis of the 1-position of 2-KG esters due to its increased leaving group potential was also described in the above mentioned Zengeya et al. reference (Org Lett 2015, 10.1021/acs.orglett.5b00737) and may have some discussion points useful for the revision.

We apologize for excluding this reference in our original manuscript. We have cited this work (ref #30) and discussed the parallel findings for 1- versus 5- TFMB-KG analogs (see below):

Lines 140-145: “5MKG, generated by hydrolysis of the ester group proximal to the α -ketone, was ~16-fold more abundant than 1MKG (Fig. 2c). This observation supports the broader conclusion that α -ketoesters are more labile. Indeed, Zengeya et al. showed that regiospecific 1-TFMB-KG esters were significantly more susceptible to non-enzymatic hydrolysis relative to 5-TFMB-KG esters, owing to the increased acidity and leaving group potential of the 1-carboxylate of KG³⁰.”

7. One part of the paper manuscript I found a little counterintuitive was the differential effects of ketoacid esters (acid pKa~2.5) versus traditional esters (acid pKa ~4.5) on media pH. At physiological pHs both of the released acids would be >90% deprotonated and therefore reasonably expected to contribute similarly to acidification. One explanation for the difference would be if, at 1 mM, the released acids exceeded the buffering capacity of the medium. Could the authors better explain this phenomena, incorporating a reference to the buffering capacity of the media?

We agree that this point and the supporting data were not well described in the initial manuscript. Surprisingly, we found that dimethyl-succinate did not hydrolyze significantly over 72 hours compared to α -ketoesters, which hydrolyzed within minutes. These data were originally in the Extended Data, but we believe that they are relevant in the main figures for clarity. We have moved these data into the revised main figures (Fig. 4e) and have included the following text changes to improve clarity (see below):

Lines 228-238: “To determine if the α -ketone carboxylic acid contributes to significant proton release, we conducted cell-free acidification experiments using structurally-similar dicarboxylates that lack an α -keto group, including dimethyl-glutarate (DM-G) and dimethyl-succinate (DM-S) (Fig. 4d). The pKa of succinic acid (C1 pKa ~4.2; C4 pKa ~5.6) and glutaric acid (C1 pKa ~4.3; C5 pKa ~5.4) are well below physiologic pH and expected to exist mainly in the deprotonated form similar to KG⁴⁹. However, injection of either DM-G or DM-S did not lead to significant acidification of media over 60 minutes relative to vehicle (Fig. 4c; Extended Data Fig. 4b). Unlike DMKG, we did not observe significant cell-free hydrolysis of DM-S to succinic acid after 72 hours by GC-MS, suggesting that hydrolysis and subsequent deprotonation may be too slow to exceed the media buffering capacity over this time frame (Fig. 4e).”

Note that our pH monitoring experiments were done in DMEM containing a HEPES buffer (at 5 mM) given the lack of CO₂ used in these assays and to match conditions used in other assays (ECAR, OCR, mass spec uptake assays). We have included the buffering capacity of this media in the results and revised methods (see below, also Reviewer 3 point 6):

Lines 216-220: “To better understand the hydrolysis mechanism and potential KG-independent effects that DMKG or 1-octyl-KG may have on cellular metabolism, we monitored pH dynamically following acute injection in cell-free conditions in the presence of a HEPES buffer, similar to previous OCR and ECAR measurement conditions.”

Lines 678-689 (methods): “XF assay media was prepared by adding 20 mM of D-glucose and 2 mM of L-glutamine to XF base medium, which contains HEPES (5 mM) (Agilent). Prior to assay, 8988T cells were washed with XF assay media twice and cells were incubated in a non-CO₂ incubator at 37°C for ~45-60 minutes in 150 µl of assay media. Injection port A was loaded with 25 µl of assay media containing 7 mM (7x concentration) of metabolite or esterified analogs or methyl acetate as a vehicle control. Basal ECAR and OCR was established for ~18 minutes followed by injection of port A. Dynamics of glycolytic and respiratory metabolism following the ~60-minute stimulation was assayed by sequential injection of oligomycin (1 µM), FCCP (0.5 µM), and rotenone/antimycin (1 µM/1 µM). Cell-free acidification was assayed alongside cell-based measurements following similar protocol but excluding cells. Both cell-free and bioenergetic (OCR/ECAR) measurements were made using a modified DMEM containing a HEPES buffer (5 mM).”

9. A recent study by Swain et al. described the distinct properties of itaconate and itaconate esters that have been commonly used in immunometabolism studies. There is some resonance with the authors findings, in that Swain et al. found non-esterified itaconate was efficiently delivered and that itaconate esters have unintended consequences. It would be great if a link could be drawn between these two studies in the discussion, and I am sure the other authors would appreciate it too.

We agree that this nice study should be further discussed (Swain et al., ref #34). We included this reference in the results section in the original manuscript, but we agree that additional discussion around the unique properties of itaconate esters would benefit the discussion (see below).

Lines 483-496: “Itaconate, a derivative of cis-aconitate produced by cis-aconitate decarboxylase (CAD), is a well-established immuno-modulatory metabolite and endogenous succinate dehydrogenase (SDH) inhibitor⁸⁴⁻⁸⁶. Esterified derivatives of itaconate (e.g., dimethyl-itaconate, 4-octyl-itaconate, 4-monoethyl-itaconate) are commonly used because of presumed poor cellular delivery of endogenous itaconate. However, several studies have shown that unmodified itaconate can be taken up by several cell types, including immune, cancer, brain, and adipocytes^{34,85,87-89}. Furthermore, itaconate esters can contribute to divergent cellular phenotypes as a result of differing analogs, incubation timing, and cellular contexts^{84,90,91}; and only recently have itaconate and its analogs been comparatively examined³⁴. These paradoxical effects and our data provide compelling evidence that α -ketoesters should be used with caution as surrogates for their respective metabolites (e.g., pyruvate, KG), and phenotypes described using DMKG, methyl-pyruvate, or other α -ketoesters should be confirmed using non-esterified metabolites, ester-derived alcohol(s), and/or orthogonal approaches.”

Reviewer #3 (Remarks to the Author):

Review of paper by Parker et al.

The authors explore the effects of aKG esters versus “pure” analogs on cellular metabolism and acidification. They demonstrate that rapid hydrolysis occurs in cell media independent of esterases and induce significant aKG-independent effects on cellular metabolism. The present study is timely and highlights important differences between cell-permeable esterified analogs and their parent compounds, but further experimentation is required to support some of the findings.

We thank the reviewer for referring to our study as “timely”, and we have added new data to address the reviewer’s concerns (see below).

1. Spontaneous hydrolysis should be examined over a shorter time course. The earliest time point examined is 24 h. How quickly does hydrolysis occur?

In cell-free conditions, the earliest time point assayed was 24 hours. In our cell-based assays, we see evidence of hydrolysis as early as 3 hours but cannot rule out a contribution of intracellular de-esterification (Reviewer Fig. 3A). To address this, we have conducted additional hydrolysis time points using DMKG and 1-octyl-KG and observe rapid hydrolysis as early as 5 minutes for both esters (Reviewer Fig. 3B, C). At early time points, similar to our original data at 24 hours, DMKG exists primarily as the mono-ester form (Reviewer Fig. 3A). We have included these new data in the revised manuscript (see below).

Reviewer Figure 3. (A) Intracellular KG, 1MKG, and 5MKG levels in 8988T cells treated with 1 mM of KG or DMKG relative to vehicle for 3 hours. (B & C) Cell-free hydrolysis of DMKG or octyl-KG over 6-hours.

Lines 75-81: “After 72 hours, the majority (>50%) of DMKG or 1-octyl-KG were detected as KG suggesting that double or single hydrolysis, respectively, can occur spontaneously and independently of cellular esterases (Fig. 1a, b). Hydrolysis kinetics were distinct between DMKG and 1-octyl-KG. KG derived from complete mono-hydrolysis of 1-octyl-KG was observed after 24 hours, whereas a gradual increase in KG release from double hydrolysis of DMKG was observed over the 72-hour time course (Fig. 1c). Hydrolysis of 1-octyl-KG occurred as early as 5 minutes in follow-up short term assays (Extended Data Fig. 1a).”

Lines 127-130: “Integration of this peak revealed that DMKG rapidly hydrolyzes to the MKG within 24 hours, and as early as 5 minutes, followed by relatively slow hydrolysis to KG, agreeing with our previous measurements of KG release from DMKG (Fig. 2b; Extended Data Fig. 2c).”

2. Similarly, intracellular aKG levels should be assessed at earlier time points. Do aKG and

ester analogs increase intracellular aKG at the same rate? This should be assessed over the same time course as the functional seahorse experiments. It is important to determine if the metabolic effects are observed because the ester analog generates intracellular aKG on a faster time course to pure aKG.

Our data in 8988T acutely treated with KG or DMKG for 3 hours suggests that KG accumulation is different between esterified and pure KG. In these studies, KG accumulation was significantly lower for DMKG despite given an equivalent concentration (Reviewer Figure 4A). However, in these data we also observed a significant accumulation of intracellular mono-methyl-KG (Reviewer Figure 4A), suggesting that the complete double hydrolysis of DMKG to KG is not instantaneous and DMKG exists as KG and mono-methyl-KG in cells (see below).

Furthermore, our new data (above, point #1) suggest that mono-hydrolysis occurs over minutes which would likely be on a similar time scale to endogenous KG import. Thus, KG likely generates higher intracellular concentrations of KG, and our new data suggest that DMKG/mono-methyl-KG contributes directly to other metabolic pathways that extracellular KG does not (e.g., mitochondrial TCA cycle) (Reviewer Figure 4B, C). Given the technical complexities and transience of hydrolysis, uptake, and compartmentalization we believe these data are sufficient to suggest that the metabolic effects between KG and DMKG are functionally distinct; however, as reviewer #2 points out esterified analogs may outperform endogenous KG at lower concentrations which requires further study (see below).

Reviewer Figure 4. (A) Intracellular KG and mono-methylated-KG (1- or 5-) levels in 8988T cells treated with vehicle, KG, or DMKG for 3 hours. (B) $^{13}\text{C}_5$ -glutamine stable-isotope dilution of TCA intermediates in AsPC-1, 8988T, and 8988S cells treated with vehicle, KG, or DMKG for 24 hours. (C) Mono-methylated succinate (M-Suc) and fumarate (M-Fum) suggests mitochondrial import and metabolism of mono-methyl-KG by TCA cycle enzymes.

Lines 145-151: “We also observed a significant accumulation of intracellular KG in both KG- and DMKG- treated cells after 3 hours (Fig. 2c). However, DMKG treatment resulted in significantly less intracellular KG relative to KG treatment, because DMKG hydrolyzed to relatively equal amounts of KG and 5-methyl-KG after 3 hours (Fig. 2c). Notably, a peak corresponding to the theoretical m/z ([M-H]: 173.045 m/z) of DMKG was not detected above baseline in DMKG-treated

cells, suggesting that DMKG may completely hydrolyze to KG or mono-esterified KG within this time frame."

Lines 440-455: "Further, it is not well understood how physiological sources of KG may influence metabolism and/or dioxygenase activity. Plasma and cerebrospinal fluid levels of KG in healthy children and adults are ~10 μM and ~5 μM , respectively, and elevated plasma KG levels (~20-60 μM) have been reported in diabetic or pyruvate carboxylase deficient children^{70,71}. We use a super-physiological concentration of 1 mM in this study to represent the concentrations of DMKG and other analogs used in published studies, which typically range from 1-4 mM up to 32 mM. However, these high concentrations may enrich for negative characteristics associated with KG esters, including acidification and effects induced by monoesters and/or the released alcohol(s). Minimizing analog concentrations may be preferred to reduce these impacts and to better match physiological conditions. It is possible that esterified analogs may outperform endogenous KG uptake at lower concentrations and/or have distinct biological effects depending on the context and mechanism of KG uptake. Previous studies using TFMB-, octyl-, and/or dimethyl- KG analogs report opposing effects on HIF1 α stabilization—via modulation of the activity of prolyl hydroxylases (PHDs)—in different cellular contexts, conditions, and KG analog concentrations^{11,28,30,31}. Further, TFMB alcohol alone, released during hydrolysis of TFMB-KG analogs, has also been shown to stabilize HIF1 α ³⁰.

3. The effects of all compounds on cellular toxicity at these long time points should be assessed.

We have conducted a proliferation experiment in 8988T cells treated with 1 mM of KG, DMKG, and 1-octyl-KG over five days. Relative to vehicle (methyl acetate), KG and DMKG did not significantly impact proliferation, but 1-octyl-KG caused a strong decrease in proliferation in 8988T cells (Reviewer Figure 5). These results match those reported in U2OS cells (PMID: 31173576), which we have discussed further in the revised manuscript (see below).

Reviewer Figure 5. Proliferation of 8988T cells treated with vehicle (methyl acetate, MeAc) or 1 mM of KG, DMKG, or 1-octyl-KG over 5 days. Representative of two independent experiments.

Lines 194-200: "To determine whether either analog cause cellular toxicity, we cultured 8988T cells for five days with methyl acetate or 1 mM KG, DMKG, or 1-octyl-KG. No significant effect on proliferation was seen for KG or DMKG, but 1-octyl-KG caused a significant and near complete suppression of proliferation (Extended Data Fig. 3b). Cellular toxicity and decreased cellular respiration caused by 1-octyl-KG, and not DMKG nor TFMB-KG, has also been reported in U2OS human osteosarcoma cells¹⁰. These results highlight potential "off-target" effects of KG esters on cellular metabolism that may influence phenotypes associated with their use."

4. Given the vast array of literature using esterified analogs in immune cells it would be interesting to repeat these experiments in an immune cell type.

We agree and have included additional reference and discussion of the immune cell literature and additional experiments in RAW246.7 macrophages. Similar to other studies cited, we found overwhelming use of DMKG without KG controls in the immune cell literature ranging from 0.5

to 5 mM (PMID: 28783731, 28714978, 26420908, 28059703, 28813658, 30120246, 30305738). We have included these references in our revised manuscript (see below):

Lines 33-37: “From a metabolism perspective, esterification is used to bypass transport-mediated import or deliver metabolites thought to be impermeable to cells. For example, α -ketoglutarate (KG) is reportedly impermeable to cell membranes and esterified analogs, such as dimethyl- α -ketoglutarate (DMKG) and octyl- α -ketoglutarate, are used to supplement cells with KG to study its role in central carbon metabolism, epigenetics, and protein hydroxylation¹⁰⁻³¹.”

Lines 62-65: “The most commonly used cell-permeable analogs of α -ketoglutarate (KG) include mono- and di- esters such as dimethyl- α -ketoglutarate (DMKG) and octyl- α -ketoglutarate (1-octyl-KG), although other analogs such as (trifluoromethyl)benzyl α -ketoglutarate (TFMB-KG), benzyl α -ketoglutarate, and trifluorobenzyl α -ketoglutarate ethyl ester (ETaKG) have been used¹⁰⁻³¹.”

Lines 81-85: “While it has been suggested that KG is impermeable to cellular membranes, the majority of studies exclude KG as a control and rely exclusively on esterified analogs^{10,13-19,21-27,29,31}. We hypothesized that the spontaneous hydrolysis of KG esters may affect cell-permeable properties assumed for these more hydrophobic analogs, and KG itself may readily import into cells.”

As the reviewer suggests, we have repeated the KG and DMKG stable-isotope dilution studies in RAW264.7 macrophages. We selected this cell line because the culture and media conditions are similar to the other cell lines used in our study, allowing more direct comparisons. In contrast to the PDAC and MEF cell lines, RAW264.7 exhibited relatively minor KG uptake, but showed a significant accumulation of KG in DMKG-supplemented cells (Reviewer Fig. 6). These cells are the only cell line where we have observed this type of profile, which may have influenced the selective use of DMKG in macrophage and other immune cell studies. Similar to our other data, we observed a significant contribution of KG derived from DMKG hydrolysis to succinate (Reviewer Fig. 6). However, unlike our new data in PDAC cell lines, we did not see contribution of DMKG to mitochondrial TCA intermediates (Reviewer Fig. 6). These new data are included in the revised manuscript (Fig. 5g), and we have expanded our discussion of the context dependence of KG uptake (see below and reviewer 1, point 2 and reviewer 2, points 1 and 2).

Reviewer Figure 6. (A) Relative intracellular KG levels in RAW264.7 cells treated with 1 mM of KG or DMKG or methyl acetate for 24 hours measured by GC-MS. (B) Contribution of KG or DMKG to mitochondrial TCA intermediates, including succinate (suc), fumarate (fum), malate (mal), glutamate (glu), and citrate (cit).

Lines 290-303: “To determine if imported KG or DMKG could act as an anaplerotic source for mitochondrial TCA cycle metabolism, we quantified ¹³C label-dilution into downstream TCA cycle intermediates (e.g., fumarate, malate, glutamate, and citrate) in HuPT4, 8988S, 8988T, and

AsPC1 cell lines as well as RAW246.7 macrophages, given the broad usage of KG analogs in immune cell types (Fig. 5f). While we observed significant incorporation of KG into succinate pools across all cell lines except for RAW246.7 macrophages, we quantified minimal contribution into TCA cycle metabolites (0.5-2%) in all KG supplemented cells (Fig. 5g, h; Extended Data Fig. 5b). To determine if this was a cancer cell line phenomenon, we repeated the stable-isotope dilution in a non-transformed mouse embryonic fibroblast (MEF) cell line, and we observed similar import of KG and decarboxylation to succinate but minimal contribution to TCA cycle metabolites (~2-3%) (Extended Data Fig. 5c)⁵⁶. These data suggest that KG import is context-dependent and does not significantly contribute to mitochondrial metabolism in normal culture conditions. Thus, the succinate derived from imported KG is likely independent of KGDH and is extra-mitochondrial.”

Lines 388-398: “Not all of the cell types evaluated in our study were able to take up KG in its endogenous form. For example, supplemented KG did not significantly contribute to intracellular KG pools in RAW264.7 macrophages (Fig. 5g), suggesting a potential lack of KG transporter expression. Previous studies by MacKenzie et al., Tennant et al., and Zengya et al. show that HEK-293, HCT116, and HeLa cells do not significantly take up KG and require octyl- and/or TFMB- modified analogs to increase intracellular KG levels and cause biological effects^{11,28,30}. We also observed considerable heterogeneity in KG uptake across the cell lines tested (Fig. 1d), emphasizing the need to carefully quantify uptake capacity and metabolic utilization using stable-isotope labeling in each context. How KG is transported by cancer, immune, and other cell types is not well understood and warrants future study.”

5. The authors state that the acidification driven by DM-KG and 1-octyl-KG is responsible for the decrease in ECAR but only DM-KG decreases ECAR, not 1-octyl-KG. How do the authors reconcile this difference?

We agree that this point is confusing and requires clarification. Of note, 1-octyl-KG led to ~50% of the acidification that we quantified using DMKG (-300 versus -500 mpH). One possibility is there may be a threshold of acidification required to decrease ECAR; however, we were unfortunately unable to test this directly. As discussed in our initial manuscript, 1-octyl-KG inhibits OCR by ~30%, and this phenomenon was not seen in other conditions. This decrease in OCR may lead to a compensatory increase in glycolytic metabolism, which may counteract an acid-induced inhibition of ECAR. Because we were not able to directly test whether the magnitude of acidification observed is sufficient to decrease ECAR, we have adjusted our discussion of these data (see below).

Lines 7-10: “Esterified KG analogs exhibited spurious KG-independent effects on cellular metabolism, including extracellular acidification, arising from rapid hydrolysis and de-protonation of α -ketoesters, and significant analog-specific inhibitory effects on glycolysis or mitochondrial respiration.”

Lines 216-223: “To better understand the hydrolysis mechanism and potential KG-independent effects that DMKG or 1-octyl-KG may have on cellular metabolism, we monitored pH dynamically following acute injection in cell-free conditions in the presence of a HEPES buffer, similar to previous OCR and ECAR measurement conditions. Injection of either DMKG or 1-octyl-KG significantly acidified culture media by ~500 and ~300 mpH, respectively (Fig. 4b). Injection of KG did not illicit a similar effect on media pH, suggesting that proton release from α -ketoglutaric acid produced during hydrolysis of KG analogs is likely the source of extracellular acidification (Fig. 4b, c).”

Lines 457-476: “Proton-coupled lactate secretion through the SLC16A family of monocarboxylate transporters (MCTs) is an important component of aerobic glycolysis⁷². Because the MCTs are

passive transporters, the accumulation of protons and/or lactate in the extracellular environment is expected to inhibit MCT activity, and consequently, glycolysis-induced extracellular acidification rate⁷³. Although DMKG, 1-octyl-KG, and M-Pyr analogs caused significant acidification, only DMKG and M-Pyr treatments significantly inhibited ECAR (Fig. 3c, d, Extended Data Fig. 4d, e). A low pH threshold may be necessary to inhibit MCT and/or glycolytic activity, and/or the decrease in OCR specifically in 1-octyl-KG treated cells may drive a compensatory increase in glycolysis, convoluting the cellular response. In contrast, the increase in OCR from pyruvate or M-Pyr supplementation likely contributes to the compensatory decrease in ECAR observed in both conditions (Extended Data Fig. 4d, e). However, the pH threshold and/or metabolic effects of KG, pyruvate, methanol, octanol, and/or mono-methyl-KG that contribute to glycolysis inhibition warrants future study. The acidification was relatively modest in cells treated with 1 mM of DMKG, 1-octyl-KG, or M-Pyr but will likely scale with increasing concentrations. For comparison, the extracellular pH_e of tumors typically ranges from 6.5 to 6.9⁷⁴. Furthermore, extracellular acidification as low as 6.0 can occur during inflammation, which leads to direct effects on the metabolism and differentiation state of immune cells recruited to these acidified microenvironments⁷⁵. Thus, although we do not know if this magnitude of acidification is directly responsible for glycolysis inhibition in this context, the impacts that KG analogs have on glycolysis and respiration occur independently of KG.”

6. The acidification assay should be performed in a cellular system, which would likely be buffered. Do these compounds still decrease pH in cells? Or across the cell membrane? There are established microscopy methods to measure pH in live cells.

Note that our pH monitoring experiments were done in DMEM containing a HEPES buffer (at 5 mM) given the lack of CO₂ used in these assays and to match conditions used in other assays (ECAR, OCR, mass spec uptake assays). Our data suggest that the acidification is significant enough to exceed the buffering capacity of the media in contrast to dimethyl-succinate, which hydrolyzed slowly and did not contribute to acidification. We have included the buffering capacity of this media in the results and revised methods (see below, also Reviewer 2 point 7):

Lines 216-220: “To better understand the hydrolysis mechanism and potential KG-independent effects that DMKG or 1-octyl-KG may have on cellular metabolism, we monitored pH dynamically following acute injection in cell-free conditions in the presence of a HEPES buffer, similar to previous OCR and ECAR measurement conditions.”

Lines 228-238: “To determine if the α -ketone carboxylic acid contributes to significant proton release, we conducted cell-free acidification experiments using structurally-similar dicarboxylates that lack an α -keto group, including dimethyl-glutarate (DM-G) and dimethyl-succinate (DM-S) (Fig. 4d). The pKa of succinic acid (C1 pKa ~4.2; C4 pKa ~5.6) and glutaric acid (C1 pKa ~4.3; C5 pKa ~5.4) are well below physiologic pH and expected to exist mainly in the deprotonated form similar to KG⁴⁹. However, injection of either DM-G or DM-S did not lead to significant acidification of media over 60 minutes relative to vehicle (Fig. 4c; Extended Data Fig. 4b). Unlike DMKG, we did not observe significant cell-free hydrolysis of DM-S to succinic acid after 72 hours by GC-MS, suggesting that hydrolysis and subsequent deprotonation may be too slow to exceed the media buffering capacity over this time frame (Fig. 4e).”

Lines 678-689 (methods): “XF assay media was prepared by adding 20 mM of D-glucose and 2 mM of L-glutamine to XF base medium, which contains HEPES (5 mM) (Agilent). Prior to assay, 8988T cells were washed with XF assay media twice and cells were incubated in a non-CO₂ incubator at 37°C for ~45-60 minutes in 150 μ l of assay media. Injection port A was loaded with 25 μ l of assay media containing 7 mM (7x concentration) of metabolite or esterified analogs or methyl acetate as a vehicle control. Basal ECAR and OCR was established for ~18 minutes followed by injection of port A. Dynamics of glycolytic and respiratory metabolism following the

~60-minute stimulation was assayed by sequential injection of oligomycin (1 μ M), FCCP (0.5 μ M), and rotenone/antimycin (1 μ M/1 μ M). Cell-free acidification was assayed alongside cell-based measurements following similar protocol but excluding cells. Both cell-free and bioenergetic (OCR/ECAR) measurements were made using a modified DMEM containing a HEPES buffer (5 mM).”

We agree that DMKG and/or 5-methyl-KG that is taken up by cells may contribute to intracellular or intra-organelle acidification through further hydrolysis and de-protonation. We have quantified relative changes to intracellular pH using the pH-sensitive SNARF-5F dye in 8988T cells treated with methyl acetate or 1 mM of KG, DMKG, or 1-octyl-KG for 1-hour using live cell confocal microscopy. These studies were done in HEPES-buffered DMEM, similar to the Seahorse and cell-free acidification assays. We observed no significant decrease in intracellular pH with KG or 1-octyl-KG treatment (Reviewer Fig. 7A, B). However, DMKG treatment significantly decreased intracellular pH, consistent with our mechanism whereby hydrolysis of 5MKG would produce α -ketoglutaric acid that would be expected to deprotonate at a pH_i > 7.4 commonly observed in cancer cell lines. However, to quantify the absolute magnitude of this intracellular pH decrease and to determine whether intracellular 5MKG hydrolysis causes this change would require a significant amount of optimization and additional experimentation that we were unable to complete in a reasonable timeframe. Thus, while these data are suggestive that we see similar intracellular pH changes as demonstrated in our cell-free systems, they are not conclusive enough to include in the revised manuscript. We hope the reviewer will appreciate that our data continues to strongly support our proposed mechanism and future detailed studies will be needed to confirm these findings. We have now expanded our discussion of the acidification results (see above point 5, also below).

Reviewer Figure 7. DMKG treatment decreases intracellular pH. (A) Relative fluorescence intensity of 8988T cells pre-loaded with SNARF-5F and treated with methyl acetate or 1 mM of KG, DMKG, or 1-octyl-KG for 1 hour before live-cell confocal microscopy; excitation 488 nm, emission 640 nm. Decreasing intensity indicates decreasing intracellular pH. Quantification of 50 cells (n=50) across four randomly acquired frames. (B) Representative fluorescent images.

Lines 457-476: “Proton-coupled lactate secretion through the SLC16A family of monocarboxylate transporters (MCTs) is an important component of aerobic glycolysis⁷². Because the MCTs are passive transporters, the accumulation of protons and/or lactate in the extracellular environment is expected to inhibit MCT activity, and consequently, glycolysis-induced extracellular acidification rate⁷³. Although DMKG, 1-octyl-KG, and M-Pyr analogs caused significant acidification, only DMKG and M-Pyr treatments significantly inhibited ECAR (Fig. 3c, d, Extended Data Fig. 4d, e). A low pH threshold may be necessary to inhibit MCT and/or glycolytic activity, and/or the decrease in OCR specifically in 1-octyl-KG treated cells may drive a compensatory increase in glycolysis, convoluting the cellular response. In contrast, the increase in OCR from pyruvate or M-Pyr supplementation likely contributes to the compensatory decrease in ECAR observed in both conditions (Extended Data Fig. 4d, e). However, the pH threshold and/or metabolic effects of KG, pyruvate, methanol, octanol, and/or mono-methyl-KG that contribute to glycolysis inhibition warrants future study. The acidification was relatively modest in cells treated with 1 mM of DMKG,

1-octyl-KG, or M-Pyr but will likely scale with increasing concentrations. For comparison, the extracellular pH_e of tumors typically ranges from 6.5 to 6.9⁷⁴. Furthermore, extracellular acidification as low as 6.0 can occur during inflammation, which leads to direct effects on the metabolism and differentiation state of immune cells recruited to these acidified microenvironments⁷⁵. Thus, although we do not know if this magnitude of acidification is directly responsible for glycolysis inhibition in this context, the impacts that KG analogs have on glycolysis and respiration occur independently of KG."

7. How does DMKG affect TCA metabolism? This is unclear. These comparators need to be performed when the absolute levels of aKG derived from exogenous aKG and ester analogs are the same. The time point here was 3 h. What do the absolute levels of aKG look like at this time point?

As we discuss in point 2, KG levels after a 3-hour treatment with 1 mM of KG or DMKG are similar in magnitude but significantly less in DMKG-treated cells (Reviewer Fig. 8A). This is because DMKG exists as roughly similar amounts of KG and 5-methyl-KG as we discuss in the manuscript and in point 2 (Reviewer Fig. 8A). However, we observe that DMKG uniquely contributes to the TCA cycle in many cell lines, whereas extracellular KG does not (Reviewer Fig. 8B). It is difficult to understand how this might affect the TCA cycle, as the stable-isotope tracers required to understand this are diluted by contributing DMKG. However, if TCA cycle flux was the same between control and DMKG-supplemented conditions, then stable-isotope dilution should affect KG-dependent mitochondrial pathways similarly. However, we see that DMKG specifically dilutes isotopologues linked to oxidative TCA cycle activity (e.g., M+4 citrate) (Reviewer Fig. 8C, D). We do not see a significant effect on the relative activity of reductive glutamine metabolism, indicated by M+5 labeled citrate, by DMKG supplementation (Reviewer Fig. 8C, D). These data suggest that DMKG may specifically contribute to oxidative TCA cycle activity. It is difficult to determine whether supplementation increases oxidative TCA cycle flux; however, we see a significant accumulation in TCA intermediates (malate, fumarate) as well as aspartate (Reviewer Fig. 8E). Notably, these effects were unique to DMKG supplementation, providing further evidence that DMKG has unique metabolic properties relative to KG.

Reviewer Figure 8. (A) Intracellular KG and mono-methylated-KG (1- or 5-) levels in 8988T cells treated with vehicle, KG, or DMKG for 3 hours. (B) ¹³C₅-glutamine stable-isotope dilution of TCA intermediates in AsPC-1, 8988T, and 8988S cells and RAW246.7 macrophages treated with vehicle, KG, or DMKG for 24 hours. (C) Citrate labeling from ¹³C₅-glutamine in AsPC-1, 8988T, or 8988S cells treated with vehicle, KG, or DMKG for 24 hours. Unlabeled (M0), oxidatively generated citrate (M4), and reductively generated citrate (M5) are indicated. (D) Schematic of oxidative versus reductive glutamate metabolism. Red circles indicate isotopologues generated through oxidative TCA cycle metabolism; blue circles indicate isotopologues generated through reductive carboxylation. (E) Percent increase in intracellular aspartate, malate, or fumarate in AsPC-1, 8988T, or 8988S cells treated with vehicle, KG, or DMKG for 24 hours.

We have included these new data and discussion in the revised manuscript (see below).

Lines 318-334: “These data indicate that 5-methyl-KG can likely be directly metabolized within cells to form methylated TCA intermediates (Extended Data Fig. 5f); however, these features must be validated and quantified with chemical standards to determine the amount produced by DMKG treatment. These methylated intermediates may further explain the incongruity between the metabolic effects of DMKG versus KG treatment that we observed. As further support for the mitochondrial permeability of 5MKG/DMKG, we found that DMKG treatment led to a significant stable-isotope dilution of TCA metabolites beyond succinate in many of the cell lines examined, including citrate, fumarate, malate and glutamate (Fig. 5g, h). Notably, RAW246.7 macrophages did not share this phenotype, and DMKG contribution was limited to succinate in this context (Fig. 5g). Analysis of oxidative (M+4) and reductive (M+5) citrate isotopologue enrichment suggests that DMKG treatment most notably contributes to oxidative TCA cycle metabolism (e.g., dilution of M+4 citrate) (Extended Data Fig. 5i, j). Further, intracellular levels of malate, fumarate, and aspartate—which are produced by oxidative mitochondrial metabolism—were significantly and mostly elevated in DMKG-treated cells (Extended Data Fig. 5j, k). Taken together, these results

highlight that DMKG contributes to cytosolic and mitochondrial metabolism and/or dioxygenase activity, whereas extracellular KG uptake and metabolism may specifically fuel cytosolic pathways.”

Reviewers' Comments:

Reviewer #1:

Remarks to the Author:

I am satisfied.

Reviewer #2:

Remarks to the Author:

The authors have suitably addressed my comments, most of which lie on the chemical side. I am not an expert in topic the other reviewer's comments and defer to their opinion as to whether those have been satisfactorily addressed.

Jordan Meier

Reviewer #3:

Remarks to the Author:

The new experiments strengthen the findings of the paper and I believe it is now acceptable. A couple of small comments:

1. The data in Raws is very interesting and certainly suggests these cells are unique in their inability to sequester endogenous aKG. The authors suggest this is due to lack of transporter expression. Which plasma membrane transporter are they referring to?
2. If cell free hydrolysis in media alone occurs on the order of minutes how do these cells take up the deesterified form?
3. I believe this statement in line 324 is incorrect "DMKG treatment led to a significant stable-isotope dilution of TCA metabolites beyond succinate in many of the cell lines examined, including citrate, fumarate, malate and glutamate (Fig. 5g, h)." It appears there is more TCA labelling with DMKG.

We are grateful for the opportunity to submit our revised manuscript to *Nature Communications*, and we are delighted that the reviewers have found the paper to be acceptable.

We appreciate the additional constructive comments by Reviewer #3, which we have addressed in the revised manuscript. Please find below a point-by-point response (in blue). Specific changes to the manuscript that clarify the remaining questions are underlined below.

Sincerely,

Alec Kimmelman and Seth Parker

The new experiments strengthen the findings of the paper and I believe it is now acceptable.

A couple of small comments:

1. The data in Raws is very interesting and certainly suggests these cells are unique in their inability to sequester endogenous aKG. The authors suggest this is due to lack of transporter expression. Which plasma membrane transporter are they referring to?

The transporter(s) involved in endogenous aKG uptake in the cells used in this study are not known, and our data in RAWs suggest that macrophages may lack expression of these transporter(s). We share the reviewer's enthusiasm about these data and plan to identify the transporter(s) involved in aKG uptake in a future study. We have clarified this in the revised text (see below).

Lines 390-393: "Not all of the cell types evaluated in our study were able to take up KG in its endogenous form. For example, supplemented KG did not significantly contribute to intracellular KG pools in RAW264.7 macrophages (Fig. 5g), suggesting that these cells lack expression of a plasma membrane KG transporter."

Lines 399-400: "How KG is transported by cancer, immune, and other cell types and which transporter(s) are involved is not well understood and warrants future study."

2. If cell free hydrolysis in media alone occurs on the order of minutes how do these cells take up the deesterified form?

Our data support a mechanism whereby mono-methyl-KG can be taken up by cells via passive diffusion and independently of a transporter. In cells that show endogenous aKG uptake (cancer cell lines, MEFs), both KG and DMKG/5MKG accumulate to similar extents (Fig. 1). This would suggest that DMKG/5MKG can either passively diffuse into the cell or be taken up through similar transporter(s) as endogenous aKG. However, RAW macrophages did not take up endogenous aKG but did take up DMKG/5MKG. These data suggest that DMKG/5MKG can be taken up by cells that lack an aKG transporter. We have clarified this in the discussion (see below):

Lines 372-379: “Our work demonstrates that many metabolite esters, including α -ketoglutarate analogs, are vulnerable to rapid, spontaneous hydrolysis in aqueous conditions and are likely taken up by cells in metabolite form or transiently as monoesters (e.g., mono-methyl-KG). While it is often assumed that cell membranes are impermeable to KG, our data suggests that many cell lines can take up KG and likely use specific transporter(s) to facilitate its diffusion across cell membranes. Esterified KG analogs accumulated in cell lines that could take up endogenous KG and those that did not (e.g., RAW264.7 macrophages), suggesting that uptake involves KG transporter(s) and/or passive diffusion across the plasma membrane.”

3. I believe this statement in line 324 is incorrect “DMKG treatment led to a significant stable-isotope dilution of TCA metabolites beyond succinate in many of the cell lines examined, including citrate, fumarate, malate and glutamate (Fig. 5g, h).” It appears there is more TCA labelling with DMKG.

The reviewer is correct. This sentence is confusing because the data in Figure 5g and 5h are presented as the “contribution to TCA cycle intermediates” (an increasing value) quantified by the “dilution” (a decreasing value) of ^{13}C -glutamine labeled TCA intermediates. We have clarified this sentence in the revised manuscript (see below):

Lines 322-324: “As further support for the mitochondrial permeability of 5MKG/DMKG, we found that DMKG contributed to TCA metabolites beyond succinate in many of the cell lines examined, including citrate, fumarate, malate, and glutamate (Fig. 5g, h).”

Reviewers' Comments:

Reviewer #3:

Remarks to the Author:

Thank you for addressing these questions. The paper is now acceptable.